# Administration of AICAR, an AMPK Activator, Prevents and Reverses Diabetic Polyneuropathy (DPN) by Regulating Mitophagy

**DOI:** 10.3390/ijms26010080

**Published:** 2024-12-25

**Authors:** Krish Chandrasekaran, Joungil Choi, Mohammad Salimian, Ahmad F. Hedayat, James W. Russell

**Affiliations:** 1Department of Neurology, University of Maryland School of Medicine, Baltimore, MD 21201, USA; krishchandra0630@gmail.com (K.C.); joungil.choi@va.gov (J.C.); mohammad.salimian@gmail.com (M.S.); fahim.hedayat@yahoo.com (A.F.H.); 2Veterans Affairs Medical Center, Baltimore, MD 21201, USA; 3CAMC Department of Neurology, Institute for Academic Medicine, 415 Morris Street Suite 300, Charleston, WV 25301, USA; 4West Virginia University, Charleston, WV 25301, USA

**Keywords:** exercise mimetics, AICAR, AMPK, diabetic neuropathy, mitochondrial fission, mitochondrial biogenesis, lipid metabolism, glucose metabolism, nerve conduction, intraepidermal nerve fiber density

## Abstract

Diabetic peripheral neuropathy (DPN) is a common complication of diabetes in both Type 1 (T1D) and Type 2 (T2D). While there are no specific medications to prevent or treat DPN, certain strategies can help halt its progression. In T1D, maintaining tight glycemic control through insulin therapy can effectively prevent or delay the onset of DPN. However, in T2D, overall glucose control may only have a moderate impact on DPN, although exercise is clearly beneficial. Unfortunately, optimal exercise may not be feasible for many patients with DPN because of neuropathic foot pain and poor balance. Exercise has several favorable effects on health parameters, including body weight, glycemic control, lipid profile, and blood pressure. We investigated the impact of an exercise mimetic, 5-aminoimidazole-4-carboxamide ribonucleotide (AICAR), on DPN. AICAR treatment prevented or reversed experimental DPN in mouse models of both T2D and T1D. AICAR in high-fat diet (HFD-fed) mice increased the phosphorylation of AMPK in DRG neuronal extracts, and the ratio of phosphorylated AMPK to total AMPK increased by 3-fold (HFD vs. HFD+AICAR; *p* < 0.001). Phospho AMP increased the levels of dynamin-related protein 1 (*DRP1*, a mitochondrial fission marker), increased phosphorylated autophagy activating kinase 1 (ULK1) at Serine-555, and increased microtubule-associated protein light chain 3-II (LC3-II, a marker for autophagosome assembly) by 2-fold. Mitochondria isolated from DRG neurons of HFD-fed had a decrease in ADP-stimulated state 3 respiration (120 ± 20 nmol O_2_/min in HFD vs. 220 ± 20 nmol O_2_/min in control diet (CD); *p* < 0.001. Mitochondria isolated from HFD+AICAR-treated mice had increased state 3 respiration (240 ± 30 nmol O_2_/min in HFD+AICAR). However, AICAR’s protection in DPN in T2D mice was also mediated by its effects on insulin sensitivity, glucose metabolism, and lipid metabolism. Drugs that enhance AMPK phosphorylation may be beneficial in the treatment of DPN.

## 1. Introduction

While there is no specific treatment for diabetic peripheral neuropathy (DPN), aerobic exercise may slow the progression of neuropathy [1,2,3,4]. Aerobic (endurance) and resistance exercise play crucial roles in improving glycemic control and insulin sensitivity in individuals with diabetes [2,4,5,6,7,8]. However, some patients with type 2 diabetes may not participate in regular physical exercise therapy [9]. One of the critical obstacles in utilizing exercise therapy is the inability of those with DPN to participate fully in weight-bearing exercises due to discomfort and poor balance related to the neuropathy [2,3]. To address this problem, researchers have explored the concept of exercise mimetics—a substance that mimics the effects of exercise without the need for physical activity.

Exercise mediates its action via the AMP-activated protein kinase (AMPK) or 5′ AMP-activated protein kinase (EC 2.7.11.31) [10,11]. Exercise decreases the ATP/AMP ratio, leading to increased AMP levels. Increased AMP promotes the phosphorylation of AMPK. Phosphorylated AMPK then promotes adenosine triphosphate (ATP) production by inhibiting ATP-consumption pathways and accelerating ATP-generation pathways. AMPK is thus acting like a ‘fuel gauge’ [12,13]. When ATP levels are low (indicating energy deficiency), AMPK is activated. AMPK in turn increases the expression of the PPARγ-coactivator (PGC-1α), which has been shown to be protective in preventing DPN [14]. Increased expression of PGC-1α leads to downstream activation of pathways that are critical in regulating mitochondrial function that in turn protect against DPN [15,16,17,18]. PGC-1α also stimulates GLUT4 translocation to the plasma membrane, increases glucose uptake, and reduces fatty acid levels, which in turn improves glycemic control. AMP regulates various pathways related to lipid metabolism and mitochondrial function. Additionally, AMPK regulates mitochondrial dynamics (fission and fusion) and their function [10,19,20,21].

Some medications used for the treatment of diabetes, such as thiazolidinediones and metformin, are mild activators of AMPK. Metformin activates AMPK by inhibiting mitochondrial function [22,23]. In contrast, 5-aminoimidazole-4-carboxamide ribonucleotide (AICAR) is a potent AMP analog, produced naturally by the body. AICAR significantly improves metabolic control in obese Zucker rats [24] and mimics the effect of exercise in mice [25]. However, nothing is known about the effect of AICAR on neuropathy. By addressing multiple risk factors, including dyslipidemia, insulin sensitivity, and glycemic control, AICAR shows promise in ameliorating DPN. This research opens possibilities for novel treatments that harness the power of AMPK.

## 2. Results

### 2.1. A High-Fat Diet (HFD) Decreases Neuronal AMPK Phosphorylation and AICAR Increased AMPK Phosphorylation in HFD-Fed Mice

Our proposed working model is that administration of AICAR can ameliorate peripheral sensory neuropathy in both HFD-induced type 2 and streptozotocin (STZ)—induced type 1 diabetic mice by regulating mitochondrial function via AMPK phosphorylation in both neuronal and non-neuronal tissues, thereby alleviating a diabetes-induced energy metabolic failure (Figure 1). To test this, AICAR was administered to control diet (CD-fed) and HFD-fed mice for 4 months. AICAR was dissolved in DMSO (5%)/alcohol (5%)/saline (90%) and 100 µL of this solution was administered subcutaneously for 3 days in a week (Monday, Wednesday, and Friday) at a dose of 500 mg/kg. The vehicle, DMSO (5%)/alcohol (5%)/saline (90%) was administered to CD- and HFD-fed mice for 4 months. Thus, there were four groups, CD (4 months), CD+AICAR (4 months), HFD (4 months), and HFD+AICAR (4 months). Baseline nerve conduction studies (NCS) and Von Frey monofilament studies were not statistically different between the mice randomized to each of the groups. At the end of 4 months, NCS were performed, mice were euthanized, and blood chemistry was obtained in the four groups of mice. Dorsal root ganglion (DRG) neurons were harvested. Protein extracts from DRG neurons were prepared, and Western blots were completed. The effect of AICAR administration on the phosphorylation of AMPK is shown in Figure 1. The ratio of AMPK phosphorylated at threonine residue 172 to total AMPK was constant in DRG neuronal extracts from CD and CD+AICAR-treated mice. On the other hand, feeding mice with an HFD caused a decrease in phosphorylation of AMPK in DRG protein extracts (Figure 2). Administration of AICAR to HFD-fed mice increased the phosphorylation of AMPK, and the ratio of phosphorylated AMPK to total AMPK increased by 3-fold (HFD vs. HFD+AICAR; *p* < 0.001).

### 2.2. Phosphorylated AMPK Promoted Mitochondrial Turnover

To correlate how AMPK phosphorylation corresponds with mitochondrial function, mitochondria were isolated from small amounts of neuronal samples following the procedure described in the Materials and Methods section. Western blots were performed on mitochondrial protein extracts to quantify mitochondrial fission/fusion proteins. The results are shown in Figure 3. In AICAR-treated samples, there was a significant increase in the levels of dynamin-related protein 1 (*DRP1*, a mitochondrial fission marker), an increase in phosphorylated autophagy activating kinase 1 (ULK1) at Serine-555, and increased microtubule-associated protein, light chain 3-II (LC3-II, a marker for autophagosome assembly). There was also an increase in the levels of the same proteins in HFD-fed neuronal tissue samples. We interpret these results as suggesting that feeding mice with an HFD does induce a physiological response in neurons to eliminate damaged mitochondria via mitophagy. Furthermore, AICAR treatment enhances this effect to protect mitochondria against HFD-induced oxidative stress.

### 2.3. AICAR Promoted Neuronal Mitochondrial State 3 Respiration

To determine if AICAR-induced mitochondrial fission in HFD-fed mice regulates mitochondrial respiration, we assessed the bioenergetics profile of mitochondria isolated from DRG neurons of CD, CD+AICAR, HFD, and HFD+AICAR mice. The oxygen consumption rate was measured using the Seahorse Biosciences analyzer (Figure 4). Basal, ADP-stimulated (state 3), and oligomycin-sensitive (state 4) respiration were measured with serial additions of 10 mM glutamate and 2 mM malate (state 2), 2.5 mM ADP (state 3), and 1.5 mM oligomycin (state 4_0_). Respiratory control ratios (state 3 to state 4_0_) were calculated. Ratios greater than 4 have been routinely observed in isolated neuronal mitochondria from WT mice. Exogenous cytochrome c did not enhance mitochondrial respiration in these preparations, indicating minimal damage to the outer mitochondrial membrane during isolation procedures. The results showed similar basal and state 4 respiration among the four samples (~53 ± 5 nmol O_2_/min/mg protein). The significant difference that was observed was that an HFD caused a decrease in ADP-stimulated, state 3 respiration compared to the other three samples (120 ± 20 nmol O_2_/min in HFD vs. 220 ± 20 nmol O_2_/min in CD and CD+AICAR samples; *p* <0.001). Mitochondria isolated from HFD+AICAR-treated mice had increased state 3 respiration (240 ± 30 nmol O_2_/min in HFD+AICAR). This suggested that AICAR treatment in HFD-fed mice was able to couple the mitochondria to energy demand, namely ADP.

### 2.4. Administration of AICAR Prevented HFD-Induced Increases in Body Weight Gain, Triglyceride, Homeostatic Model Assessment for Insulin Resistance (HOMA-IR) Index, and Non-Esterified Fatty Acids (NEFA) Levels in Mice

Several studies have shown that the main effects of AICAR occur in peripheral tissues, particularly in muscle. Muscles are a major consumer of glucose and regulator of lipid metabolism. To determine if AICAR treatment also altered glucose and lipid metabolism, body physiology and blood chemistry were completed in all four groups of mice. Significant changes were noted in the following parameters: AICAR administration to HFD-fed mice for 4 months caused a significant decrease in their body weight (from 50 ± 5 g to 42 ± 4.5 g; *p* < 0.01) and a significant decrease in triglycerides (1.4 ± 0.3 mmol/L in HFD-fed mice to 1.0 ± 0.1 mmol/L in HFD+AICAR mice; *p* < 0.01). The HFD-fed mice showed increased insulin resistance: the HOMA-IR index in HFD-fed mice was 10 ± 1 (*p* < 0.001) compared to CD-fed mice (5 ± 0.6) and to CD+AICAR-treated mice (5.1 ± 0.7). In the AICAR-treated HFD-fed mice, the HOMA-IR index was 7.3 ± 0.6; *p* < 0.01 compared to HFD-fed mice. AICAR treatment also decreased NEFA levels (HFD = 0.7 ± 0.08 mmol/L vs. HFD+AICAR = 0.4 ± 0.01 mmol/L; *p* < 0.01). The measurement of intraperitoneal-glucose tolerance test (IP-GTT; *n* = 6) showed a significant increase in the area under the curve (AUC) in HFD mice compared to CD mice and CD+AICAR-treated mice. There was a significant decrease in AUC in HFD+AICAR-treated mice compared to HFD mice (Table 1). We suggest that some of the improvements in AICAR-treated HFD-fed mice are due to their effect on reducing lipid levels, decreasing insulin resistance, better glycemic control, and resistance to an increase in body weight.

### 2.5. Subcutaneous Administration of AICAR Prevented Peripheral Neuropathy Induced by HFD

We tested if administration of AICAR would prevent HFD-induced peripheral neuropathy. AICAR was injected into HFD-fed mice for 4 months. After 4 months of HFD feeding (*n* = 6), C57BL6 mice showed a significant slowing of sciatic motor nerve conduction velocity (SMNCV, Figure 5) (from 41 ± 2 m/s with a CD to 29 ± 1.2 m/s with an HFD; *p* < 0.001), increased tail motor latency (TML, from 1.3 ± 0.07 ms in CD to 2.67 ± 0.3 ms; *p* < 0.001 with an HFD), and decreased tail sensory nerve conduction velocity (TSNCV, from 34.6 ± 0.9 m/s in CD to 29 ± 1.4 m/s in HFD; *p* < 0.001). These changes in nerve conduction velocity (NCV) in HFD-fed mice were consistent with the development of peripheral neuropathy. There was no statistically significant difference in the compound muscle action potential (CMAP) or sensory nerve action potential (SNAP) amplitudes compared between groups or compared at the start and end of the treatment. This is expected because of considerable variability in the CMAP and SNAP amplitudes obtained in nerve conduction studies (NCS) in both mice and humans, especially when interval repeat studies are performed [26,27,28]. The CMAP sciatic amplitudes (mV) after 4 months of treatment were as follows: CD: 7.28 ± 2.42 (SEM); CD+AICAR: 5.53 ± 1.65; HFD: 3.60 ± 0.98; HFD+AICAR: 3.602 ± 0.979. The CMAP tail amplitudes (mV) after 4 months of treatment were as follows: CD: 1.58 ± 0.15; CD+AICAR: 1.71 ± 0.09; HFD: 1.22 ± 0.18; HFD+AICAR: 1.57 ± 0.37. The SNAP tail amplitudes (μV) after 4 months of treatment were as follows: CD: 93.45 ± 9.20; CD+AICAR: 103.21 ± 15.01; HFD: 126.51 ± 26.39; HFD+AICAR: 95.18 ± 10.09.

After 4 months on an HFD, there was a significant decrease in the Von Frey paw withdrawal threshold in HFD compared to CD mice (CD = 1.42 ± 0.22 g vs. HFD = 0.64 ± 0.1 g; *p* < 0.001, consistent with the development of tactile allodynia). In contrast, the AICAR-injected HFD-fed mice had preserved NCVs, and the velocities were comparable to CD-fed mice (Figure 5). These findings are consistent with protection against peripheral neuropathy by the administration of AICAR. HFD+AICAR mice had normal tactile allodynia at 4 months compared to CD or CD+AICAR mice. Four months after feeding mice with either a CD or an HFD with and without AICAR treatment, the mice were euthanized, and paw skins were examined for IENFD (Figure 5). Skin biopsies showed a significant decrease in the IENFD of HFD mice (14.7 ± 2.35 fibers/mm) compared to CD mice (29 ± 4 fibers/mm; *p* < 0.001). On the other hand, in AICAR-treated mice fed with a CD or an HFD, the IENFD was the same as in CD mice (HFD+AICAR = 31 ± 3 fibers/mm). These results suggest that the administration of AICAR protected against HFD-induced peripheral neuropathy.

### 2.6. Subcutaneous Administration of AICAR Reversed HFD-Induced Peripheral Neuropathy in Mice

We tested whether the administration of AICAR could reverse DPN in an HFD-induced T2D model of peripheral neuropathy. C57BL6 mice (*n* = 8) were fed with an HFD for 2 months, and NCS were performed (Figure 6). After 2 months on an HFD, there was a significant decrease in the SMNCV, TSNCV, and Von Frey paw withdrawal threshold and an increase in the TML in the HFD mice compared to the CD mice that persisted up to 4 months and is consistent with developing DPN (Figure 6). At 2 months, AICAR was then administered to CD- and HFD-fed mice for an additional 2 months. At the end of 4 months, NCS were performed in all four groups. The results are shown in Figure 6. Administration of AICAR for a further 2 months in mice originally on an HFD alone for 2 months resulted in reversal of the NCVs and Von Frey thresholds, consistent with reversal of DPN (Figure 6). Four months after feeding mice with either a CD or an HFD, the mice were euthanized, and the paw skins were examined for the IENFD. Skin biopsies showed a significant decrease in the IENFD in the HFD mice (14.5 ± 2.5 fibers/mm) compared to the CD mice (29.6 ± 4.4 fibers/mm; *p* < 0.001). In contrast, in AICAR + HFD-fed mice, the IENFD were restored to CD mice levels (HFD+AICAR = 30.8 ± 3.2 fibers/mm). These results suggest that the administration of AICAR reversed the HFD-induced peripheral neuropathy.

### 2.7. Subcutaneous Administration of AICAR Reversed Type 1 Diabetic Peripheral Neuropathy Induced Via Streptozotocin (STZ) in Mice

In this experiment, we investigated whether AICAR would reverse DPN in a T1D model of STZ diabetic mice. To test this hypothesis, we purchased 3-month-old C57Bl6 STZ-induced diabetic mice (*n* = 20) and non-diabetic C57Bl6 mice (*n* = 12). Baseline neuropathy measurements were taken in non-diabetic and STZ-induced diabetic mice immediately after STZ induction (Figure 7, time point 0), and then all mice were fed a control diet (Harlan Teklad) for 1 month. At the end of 1 month, neuropathy measurements were made in all mice. Once neuropathy developed, based on measurements of the NCS, the diabetic mice were split into two groups: one group was administered AICAR in DMSO (5%)/alcohol (5%)/saline (90%) vehicle daily subcutaneously (*sc*) at a dose of 500 mg/kg for an additional 2 months (Group #4), and another group of diabetic mice was administered daily *sc* with the vehicle (Group #3) for an additional 2 months. Non-diabetic mice, likewise, were split into two groups. One group of mice was administered AICAR in vehicle (Group #2), and another group was administered the vehicle alone for 2 months (Group #1). At 3 months, nerve conduction studies and MA were measured in all four groups of mice. The results showed that at 1 month after STZ-induced diabetes, compared to the non-diabetic mice, the STZ mice had a significantly slower sciatic motor nerve conduction velocity (SMNCV) (from 43 ± 6 m/s in non-diabetic mice to 26.6 ± 7 m/s in the STZ mice; *p* < 0.001); an increased TML (from 1.12 ± 0.2 msec to 2.17 ± 0.09 msec; *p* < 0.001); decreased TSNCV (from 40.7 ± 2.6 m/s to 26.4 ± 3 m/s; *p* < 0.001), and had developed MA (Figure 6). However, the administration of AICAR to the STZ-induced diabetic mice for a further 2 months resulted in normal tactile allodynia at 3 months, while the von Frey paw withdrawal threshold was still decreased in the STZ mice (Figure 7). Hind paw skin biopsies at 3 months, showed that the intraepidermal nerve fiber density (IENFD) was significantly decreased in the STZ mice compared to the non-diabetic mice. In contrast, the IENFD was higher in the STZ + AICAR mice compared to the STZ mice (STZ = 11.5 ± 1.8 fibers/mm vs. STZ + AICAR = 25.1 ± 2.9 fibers/mm; *p* < 0.001). These findings show that the administration of AICAR could effectively reverse peripheral neuropathy in STZ-induced diabetic mice.

## 3. Discussion

### 3.1. DPN Affects Peripheral Neuronal Mitochondrial Metabolism

Mitochondria play a crucial role in providing energy (ATP) for maintaining membrane potential, supporting neuronal plasticity, and preventing neuropathy [16,29,30]. Several intersecting pathways can regulate mitochondrial function in DPN, including PGC-1α, TFAM, and SIRT1 [16,17,18]. Activation of AMPK acts as a common mechanism leading to an increase in SIRT1 and PGC-1α signaling with an improvement in mitochondrial function [14,31]. This study evaluated the potential of the exercise mimetic AICAR, an AMP analog, in promoting the elimination of dysfunctional mitochondria to prevent and reverse DPN.

### 3.2. Exercise Decreases Neuropathic Pain and Improves Cutaneous Nerve Fiber Branching in Diabetic Patients

Exercise plays a crucial role in reducing neuropathic pain and improving cutaneous nerve fiber branching in diabetic patients [2,4,6,32,33,34,35]. Moderately intense aerobic and resistance exercise has been shown to significantly reduce pain and neuropathic symptoms in people with DPN [4,6,33]. Moderate aerobic exercise benefits cutaneous regeneration capacity, even in patients with metabolic syndrome [8]. Both aerobic (endurance) and resistance exercise contribute to improved glycemic control and insulin sensitivity in individuals with diabetes [11,32,36]. Thus, a therapy, such as AICAR, that functions as an exercise mimetic would be expected to prevent or reverse DPN.

### 3.3. Exercise and AICAR Activate AMPK in Non-Neural Tissues

In response to exercise, the key molecule that modulates energy metabolism in muscle is AMP-activated protein kinase (AMPK) [10,37]. AMPK directly binds adenine nucleotides and senses the cellular energy status. When the ATP-to-ADP or ATP-to-AMP ratio changes, AMPK is activated through an allosteric mechanism, stimulating its kinase activity. AICAR, an AMP analog, binds to the same site as AMP and activates AMPK [10]. Once activated, AMPK orchestrates phosphorylation of key proteins in pathways related to lipid oxidation, glycolysis, and mitochondrial homeostasis [38,39,40]. In this study, compared to HFD mice, AICAR administration to HFD-fed mice over 4 months resulted in decreased body weight, reduced triglycerides, improvement in the HOMA-IR index, lower NEFA levels, and improved glycemic control (AUC in IP-GTT). The observed improvements in neuropathy measures in AICAR-treated HFD-fed mice could result from its effects on lipid reduction, decreased insulin resistance, better glycemic control, and resistance to weight gain [39,41,42,43]. This is supported by findings in non-neuronal tissue that show AMPK enhances the concentration of glucose transporters like GLUT 1 and GLUT 4, resulting in the lowering of blood glucose levels and the influx of blood glucose into cells [44]. Furthermore, AMPK increases insulin sensitivity and decreases insulin resistance. This was observed in primary Schwann cells treated with the saturated fatty acid palmitate to mimic prediabetic conditions to cause insulin resistance, which was reversed by AICAR [45].

### 3.4. AICAR Regulates Mitochondrial Homeostasis in DRG Neurons

Mitochondrial fission and fusion are important in maintaining both oxidative and reductive biosynthesis in response to a change in the availability of nutrients and bioenergetic demand [46]. Recent research has shown that there is specialization of mitochondria determined by the presence of cristae that promotes OXPHOS, while proline synthesis promotes mitochondria lacking in cristae. Thus, mitochondrial fission and fusion are important in maintaining both oxidative and reductive biosynthesis in response to changing nutrient availability and bioenergetic demand [46]. The primary effect of AICAR administration in high-fat diet (HFD)-fed mice was the regulation of mitochondrial homeostasis [10,19,20,21,22,47,48,49,50,51,52]. AICAR activated AMP-activated protein kinase (AMPK) in DRG neurons of HFD-fed mice (Figure 2). This activation led to phosphorylation of mitochondrial fission factor (MFF), recruitment of dynamin-like protein DRP1 to mitochondria, and activation of ULK1, an upstream kinase in autophagy and mitophagy. Mitochondrial fission is essential for mitophagy (Figure 3). It allows damaged mitochondria to be selectively degraded through mitophagy pathways. These findings are consistent with observations that AICAR treatment in primary myotubes increases FOXO3 expression and the expression of the mitochondrial E3 ligase Mul1 involved in mitochondrial mitophagy [51,53]. Furthermore, deacetylation of SIRT1 can induce mitophagy through the PINK1-Parkin pathway, while activation of the AMPK-mTOR-ULK complex induces autophagic cell death [54].

Importantly, AICAR also activates the mitochondrial biogenesis pathway [55,56,57,58,59]. This involves increasing the expression of PGC-1α and SIRT1 [56]. AICAR treatment brings about mitochondrial homeostasis by eliminating dysfunctional mitochondria and synthesizing new functional mitochondria. These effects occur in various peripheral tissues. Importantly, recent research reveals that DPN decreases SIRT1 expression in rodent skin [17]. Reduced SIRT1 leads to a decrease in brain-derived neurotrophic factor (BDNF). Loss of BDNF affects Meissner corpuscles, which are mechanoreceptors expressing the BDNF receptor TrkB. AICAR-induced overexpression of SIRT1 in skin promotes Meissner corpuscle reinnervation and regeneration. Consequently, this results in a significant improvement in diabetic MA.

### 3.5. AICAR for DPN

Metformin, in high doses, is an indirect activator of AMPK and currently serves as a first-line therapy for T2D [22,47,60,61,62,63]. Interestingly, it is also proposed as a potential therapeutic treatment for CNS neurological diseases. Metformin is a relatively weak activator of AMPK. Metformin would be expected to enhance neuronal bioenergetics, promote mitophagy, support nerve repair, and reduce toxic protein aggregates in neurological conditions. However, AICAR could be an even better therapeutic agent for neurological diseases. AICAR directly activates AMPK. It is transported into cells by the nucleoside transporter ENT1. AICAR increases the phosphorylation of AMPK only in HFD-fed mice and not in CD-fed mice (see Figure 2). Feeding with an HFD decreases AMPK phosphorylation, while AICAR reverses this process by binding to the AMP site on AMPK and inducing AMPK phosphorylation. Phosphorylated AMPK, in turn, promotes mitochondrial turnover to eliminate dysfunctional mitochondria, reduces mitochondrial oxidative stress, and contributes to its neuroprotective properties.

AICAR (acadesine) has been used in human studies. AICAR has been used to treat surgical ischemia to reduce post-perfusion myocardial infarction [64] and relapsing chronic leukemia [65]. In humans, doses up to 210 mg/kg IV are well tolerated [65]. At doses up to 100 mg/kg, only mild and transient side effects are reported equally in placebo and drug groups [66]. Thus, the dose used in this study in mice would fall well within the therapeutic human equivalent dose [67]. At doses greater than 200 mg/kg, adverse effects included hyperuricemia that occurred commonly but was not clinically significant and resolved with the administration of prophylactic allopurinol [65]. Other adverse events included transient anemia and/or thrombocytopenia (not clinically significant), renal impairment, and transient infusion-related hypotension (clinically significant) [65]. Furthermore, AICAR was well tolerated in more than 4000 cardiac patients [64,66,68,69,70]. Unfortunately, the energy-promoting effects of AICAR have led to abuse in human athletes and animals involved in sports [71]. To place this in context, AICAR is a normal cellular intermediate in the human body [72] at concentrations necessary for human cellular activity, is orally active [25], but bioavailability is poor [66]. Thus, at physiological concentrations, AICAR is non-toxic in the body. However, if AICAR is abused at high, sustained doses, it would be expected to be toxic. However, there is interest in developing direct AMPK activators that may be safer. For example, DW14006, administered at a dose of 15 or 30 mg/kg by i.p. injection for 4 weeks, has been shown to ameliorate DPN by regulating mitochondrial dysfunction, oxidative stress, and inflammation [73].

### 3.6. Potential Limitations of the Study

We did not address the combined effect of exercise and AICAR. This is because the premise in this study is that many patients with DPN cannot exercise due to significant neuropathic pain and poor balance and therefore need an alternative to exercise to reduce the effect of DPN. However, exercise has been tested in mice for running endurance. In sedentary mice, 4 weeks of AICAR treatment alone induced metabolic genes and enhanced running endurance by 44% [25]. However, it was also shown in the same study that AMPK activation and exercise training synergistically increased oxidative myofibers and running endurance in adult mice [25,74]. Other potential limitations of the study include no statistically significant difference in the compound CMAP or SNAP amplitudes compared between groups or compared at the start and end of the treatment. There were statistically significant differences in the motor and sensory conduction velocities and latencies as described in the results. The increased variances in the CMAP or SNAP amplitudes have been previously recognized [26,27,28]. Variances may have been improved by increasing the number of animals tested or the number of data points in each group. Furthermore, there were no measurements of the number of innervated Meissner’s corpuscles in the paw pads. In a recent publication, we and colleagues showed that in both HFD and STZ mouse models of DPN, there was profound loss of Meissner corpuscles as well as degeneration or retraction of the Aβ sensory axons that innervate them [17].

## 4. Materials and Methods

### 4.1. CD and HFD

All animal protocols followed the National Institutes of Health (NIH) Guide for the Care and Use of Laboratory Animals and were approved by the Institutional Animal Care and Use Committee. To ensure that our results are robust, unbiased, and reproducible, the following was performed for all experiments: (1) random assignment of mice; (2) performance of behavioral studies by personnel blinded to the identity of the groups; (3) blind assessment of outcome measures; and (4) specific exclusion criteria identified prior to initiation of all experiments. Three-month-old, wild-type, C57BL6 mice were fed with a CD or an HFD. The CD (Harlan Teklad #2018) contained 6.2% fat (18% calories from fat), 18.6% protein (24% calories from protein), and 44.2% carbohydrate (58% calories from carbohydrate). The HFD (Bio-Serv #F3282) contained 36% fat (60% calories from fat), 20.5% protein (15% calories from protein), and 37.5% carbohydrate (26% calories from carbohydrate). In the prevention studies, NCS and Von Frey measurements were performed at baseline, and the animals were randomized into the treatment groups. We ensured that there was no difference between any of the test groups in the baseline measurements. Those performing the testing were blinded to the test group. Animals in the HFD group were treated with the HFD as described above for a period of 4 months. Animals treated with AICAR were given AICAR 500 mg/kg sc for the duration of the study as described in Section 4.2. NCS, Von Frey, IENFD, and tissues were collected at 4 months. In the reversal study, baseline studies were performed, and animals were randomized into the treatment groups. Baseline values were confirmed to not be statistically different between the treatment groups. Three-month-old male C57BL6 WT mice were fed with either a control diet (CD) or a high-fat diet (HFD) for 2 months. NCS and Von Frey testing was performed at baseline, 2 months, and 4 months. After confirming that consumption of the HFD for 2 months induced development of peripheral neuropathy as observed by the changes in the NCSs and MA, AICAR (as in Section 4.2 below) was administered to the CD and HFD mice at a dose of 500 mg/kg for an additional 2 months. The vehicle was administered to CD and HFD mice. NCS were performed 2 months after administration of the AICAR or the vehicle (4-month time point).

### 4.2. Diabetes Induction with STZ

Three-month-old male C57BL6 WT and streptozotocin (STZ)-induced diabetic mice were purchased from Jackson Labs. Six-week-old males were identified, weighed, and a baseline non-fasted glucose measurement was taken with a OneTouch Ultra2 or Ultra/Mini glucometer. Mice received daily IP injections of 50 mg STZ/kg body weight (C57BL/6J) for five consecutive days; age-matched controls received buffer injections. Mice were housed in disposable cages with absorbent bedding and ad libitum access to food and water while they metabolized STZ for at least 48 h after the final injection. After transfer to regular cages, mice were observed daily, then weighed and glucose-tested 7–14 days after the final injection. In addition, only healthy mice without significant changes in body weight were used.

Baseline neuropathy measurements were taken in both groups (time point 0), and then all mice were fed a control diet (Harlan Teklad #2018) for 1 month. Diabetic mice having a fasting blood glucose level of 300 mg/dL (11 mmol/L) were used (8 weeks old). At the end of the month of STZ diabetes, neuropathy measurements were performed in all mice. Once DPN developed, the diabetic mice were split into two groups: one group was administered AICAR, which was dissolved in DMSO (5%)/alcohol (5%)/saline (90%), and 100 µL of this solution was administered subcutaneously (*sc*) for 3 days in a week (Monday, Wednesday, and Friday) at a dose of 500 mg/kg for an additional 2 months (Group #4), and another group of diabetic mice were administered the vehicle, DMSO (5%)/alcohol (5%)/saline (90%) for an additional 2 months (Group #3). Non-diabetic mice, likewise, were divided into two groups. One group of mice was administered AICAR in the vehicle (Group #2), and another group was administered the vehicle alone for 2 months (Group #1). At 3 months, nerve conduction studies and MA were measured in all four groups of mice.

### 4.3. Western Blot Analysis

Mouse DRG neurons were collected, and proteins were extracted with an ice-cold extraction buffer with protease and phosphatase inhibitors [18,30]. Protein concentration was determined with a BCA kit (BCA Protein Assay Kit, P0010). Equal amounts of protein samples (30 µg) were electrophoresed on SDS-PAGE gels and transferred to PVDF membranes [30]. The membranes were blocked with 5% BSA in a TBST buffer and incubated overnight at 4°C with different primary antibodies: The sources of the various antibodies used in this study were: Cell Signaling: AMPKα Antibody #2532; Phospho-AMPKα (Thr172) Antibody #2531; β-Actin (13E5) Rabbit mAb #4970; LC3A/B (D3U4C) XP^®^ Rabbit mAb #12741; DRP1 (D6C7) Rabbit mAb #8570; Phospho-ULK1 (Ser555) (D1H4) Rabbit mAb #5869; VDAC Antibody #4866; ENT1 Polyclonal Antibody, PA5-116468 (Invitrogen, Carlsbad, CA, USA). The intensity was normalized to β-actin or VDAC. After rinsing with TBST, the membranes were incubated with a secondary antibody, peroxidase-conjugated goat anti-rabbit IgG (H+L). The membranes were developed with an advanced reagent (Bio-Rad, Hercules, CA, USA), and the protein bands were visualized with an automatic chemiluminescence apparatus (Bio-Rad, USA). The densities of the bands were determined using the Bio-Rad software 2.4.

### 4.4. Isolation of Mitochondria

Dorsal root ganglion mitochondria from mouse brain were isolated using a Percoll (Amersham Biosciences, Piscataway, NJ, USA) gradient centrifugation as described previously [14]. Briefly, DRG neurons were homogenized in ice-cold mitochondria isolation buffer (225 mmol/L mannitol, 75 mmol/L sucrose, 5 mmol/L HEPES, 1 mmol/L EGTA, 1 mg/mL fatty-acid-free bovine serum albumin [BSA], pH 7.4 at 4°C). The homogenate was centrifuged at 1300× *g* for 3 min, and the pellet was resuspended and centrifuged again at 1300× *g* for 3 min. The pooled supernatants were centrifuged at 22,200× *g* for 8 min, the crude mitochondrial pellet was resuspended in 15% Percoll, and layered on a pre-formed gradient of 40% and 24% Percoll. After centrifugation at 31,700× *g* for 8 min, the mitochondria were collected from the interface of the lower two layers, diluted with isolation medium, and centrifuged at 16,700× *g* for 10 min. The mitochondrial pellet was then resuspended to a final volume of 1.5 mL of mitochondrial storage buffer (230 mM mannitol, 70 mM sucrose, 0.02 mM EGTA, 20 mM Tris-HCl, 5 mM K_2_HPO_4_, pH 7.4) and centrifuged at 22,000× *g* for 10 min at 4 °C in a microfuge. The resulting mitochondrial pellet was resuspended in the mitochondrial storage without EGTA. Protein concentrations were determined by the BCA method.

### 4.5. Mitochondrial Respiration

Isolated mitochondria were diluted in cold 1x respiration buffer (230 mM mannitol, 70 mM sucrose, 0.02 mM EDTA, 20 mM Tris-HCl, 5 mM K2HPO4 at pH 7.4, 37 °C) and transferred to a Seahorse 24-well plate (10 µg Mt in each well) and spun at 2000× *g* for 20 min to settle the mitochondria [75,76,77]. To assess complex I-mediated respiration, malate and pyruvate were used as substrates. State 3 respiration was initiated by the addition of 40 mM ADP. Approximately 2 min later, state 3 respiration was terminated, and state 4^o^ respiration (resting) was initiated with the addition of 20 µM oligomycin, an inhibitor of the mitochondrial ATP synthase. The maximal rate of uncoupled respiration was subsequently measured by the addition of 40 µM carbonyl cyanide p-(trifluoromethoxy)phenylhydrazone (FCCP), which is a protonophore uncoupling molecule. Rates are an average of 4–6 independent experiments for complex I respiration.

### 4.6. Neuropathy Measurements

The guidelines of the European diabetic neuropathy study group of the EASD (Neurodiab) were used to test peripheral neuropathy [26]. Somedic von Frey monofilaments were used to assess MA [14,30]. Nerve conduction studies were performed as described [30]. The PGP9.5 antibody staining, in a blinded fashion, was used to measure the IENFD, as previously described [30]. The IENFD was calculated (as fibers/mm) by the number of complete baseline crossings of nerve fibers at the dermo–epidermal junction divided by the measured length of the epidermal surface using standardized validated methods [78,79].

### 4.7. Statistical Analysis

Variances from the mean in described data are standard errors of the mean (SEM) unless otherwise described. To determine the significance among the groups, the data were transformed using factorial ANOVA with a post hoc Tukey test, and then a comparison of dependent variables was performed. Individual comparisons were made using Student’s t-test, assuming unequal variances as previously described [80]. The associations between mitochondrial function and measures of neuropathy (NCVs and MA) were evaluated using Spearman correlation statistics.

## 5. Conclusions

In summary, the administration of AICAR increased phosphorylation of AMPK. Phosphorylated AMPK, in turn, activated proteins involved in the segregation of mitochondria by promoting mitochondrial fission. Administration of AICAR also improved insulin sensitivity, glucose, and lipid metabolism. These metabolic effects, in turn, protected or reversed DPN in both T1D and T2D mouse models. Drugs targeting AMPK phosphorylation, such as AICAR, may function as exercise mimetics. Humans with DPN are often disabled and cannot exercise adequately. Thus, exercise mimetics could play a crucial role as therapy in DPN, but this would require further clinical trials in humans to assess this. AICAR directs AMPK activation and transport via ENT1 and makes it an intriguing candidate for further investigation to develop isoform-specific AMPK activators in the nervous system. The neuroprotective properties of AMPK activators could be utilized in DPN, but further studies are needed in humans.

## Figures and Tables

**Figure 1 ijms-26-00080-f001:**
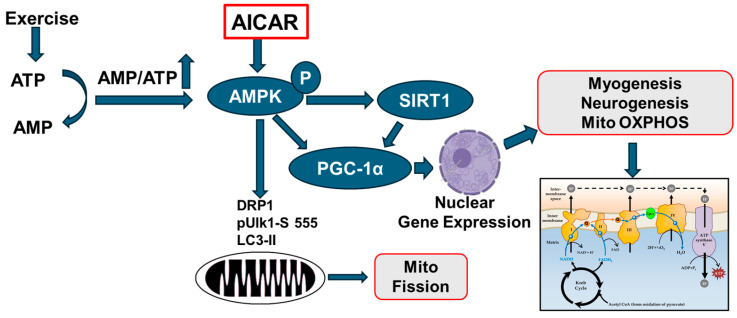
Exercise consumes a large amount of ATP, elevating the AMP-to-ATP ratio. Increased AMP binds to the enzyme AMPK and induces the phosphorylation of AMPK (Figure 1). On the other hand, **AICAR directly phosphorylates AMPK**. (1) Phosphorylated AMPK activates SIRT1/PGC1-α in the catabolic process of exercise (increased OXPHOS), resulting in decreased body weight, reduced triglycerides, improved HOMA-IR index and lower NEFA levels; (2) phosphorylated AMPK activates PGC-1α co-transcriptional complexes that initiate the overexpression of target genes to promote myogenesis, neurogenesis, and mitochondrial bioenergetics; and (3) phosphorylated AMPK led to phosphorylation of mitochondrial fission factor (MFF), recruitment of dynamin-like protein DRP1 to mitochondria, and activation of ULK1, an upstream kinase in autophagy and mitophagy. Mitochondrial fission allows damaged mitochondria to be selectively degraded through mitophagy pathways. The main points of this paper are boxed in red.

**Figure 2 ijms-26-00080-f002:**
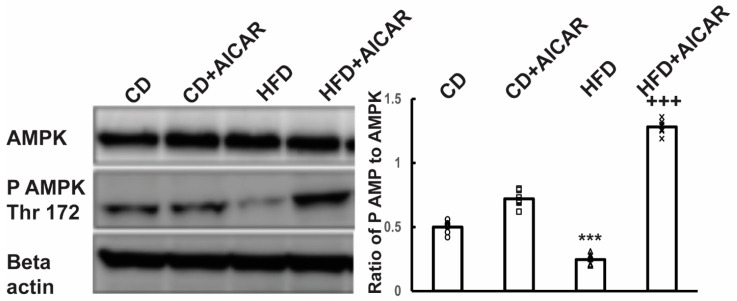
Western blot analysis of AMPK and quantification of phospho-AMPK to total AMPK (Figure 2). DRG neurons were isolated from CD (*n* = 6), CD+AICAR (*n* = 6), HFD (*n* = 6), and HFD+AICAR (*n* = 6) mice, and protein extracts were prepared. Antibodies recognize total AMPK, AMPK phosphorylated at the Thr 176 residue, and beta-actin. The levels of expression were quantified by the intensity of the bands. The ratio of pAMPK to total AMPK was calculated, and the values were analyzed by ANOVA. The significance is denoted by the following *p*-values: *** *p* < 0.001; CD or CD+AICARD vs. HFD, +++ *p* < 0.001; HFD vs. HFD+AICAR.

**Figure 3 ijms-26-00080-f003:**
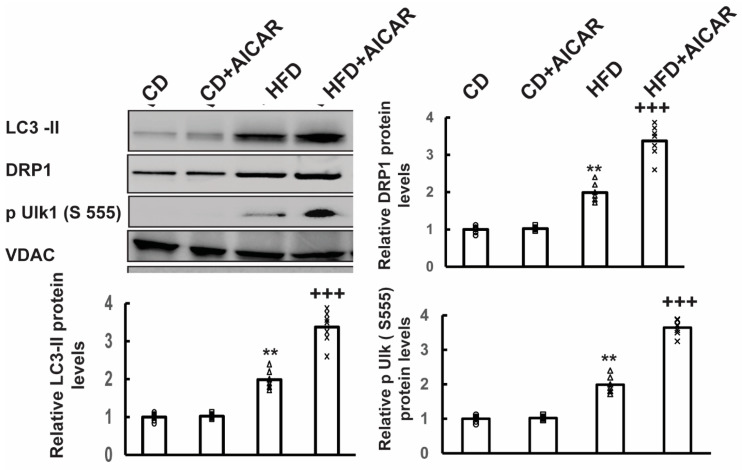
Western blot analysis of proteins involved in mitophagy in neuronal mitochondria isolated from CD (*n* = 6), CD+AICAR (*n* = 6), HFD (*n* = 6), and HFD+AICAR (*n* = 6) mice. Western blots were carried out on the protein extracts using anti-LC3, anti-DRP1, anti-phospho (S555) ULK1, and anti-VDAC. The levels of expression were quantified based on the intensity of the bands. The relative ratio was calculated, and the values were analyzed by ANOVA. The significance is denoted by the following *p*-values: ** *p* < 0.01 HFD vs. CD or CD+AICAR; +++ *p* < 0.001 HFD+AICAR vs. HFD.

**Figure 4 ijms-26-00080-f004:**
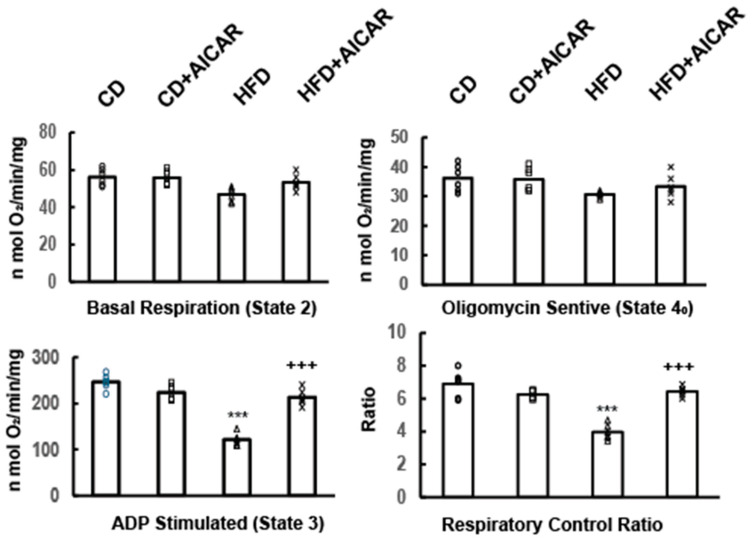
Impaired ADP-stimulated mitochondrial respiration in HFD mice was prevented by AICAR treatment. Oxygen consumption rate (OCR) was measured in the presence of complex I substrates (malate/glutamate) with the subsequent and sequential addition of ADP, oligomycin, and rotenone + antimycin A to mitochondria to measure state 2 (basal), state 3 (ADP-stimulated), state 4_0_ (oligomycin-sensitive), and FCCP-induced respiration rates were measured. ADP-stimulated respiration was significantly decreased in neuronal mitochondria from HFD-fed mice compared to CD and CD+AICAR-treated mice. Administration of AICAR to HFD-fed mice significantly increased ADP-stimulated respiration. The respiratory control ratio (RCR) was calculated. *** *p* < 0.001 HFD vs. CD, CD+AICAR. +++ *p* < 0.001 HFD vs. HFD+AICAR.

**Figure 5 ijms-26-00080-f005:**
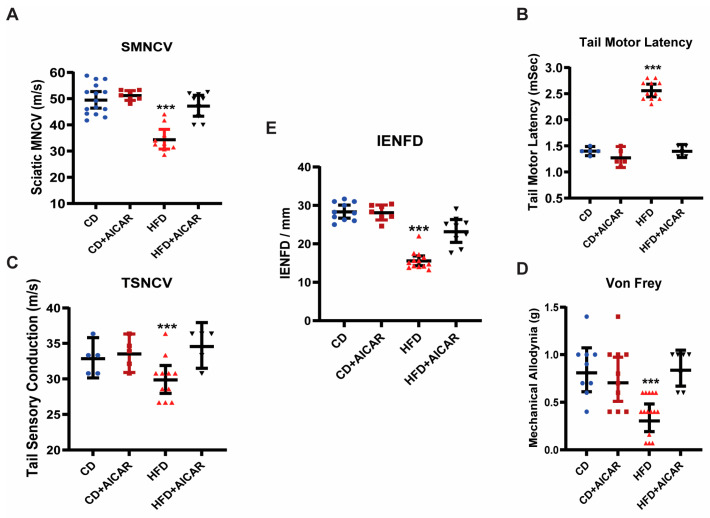
**AICAR prevented HFD-induced neuropathy in C57Bl6 mice (*n* = 6 to 8 per group).** WT C57BL6 mice were randomly assigned to four groups of six to eight mice per group. Group #1: CD; Group #2: CD+AICAR (500 mg/kg); Group #3: HFD; and Group #4: HFD+AICAR (500 mg/kg). Placebo or AICAR was administered for 4 months to CD and HFD mice. Mice were tested for the following parameters: SMNCV (**A**), TML (**B**), TSNCV (**C**), mechanical allodynia (MA) by Von Frey filament paw withdrawal threshold (**D**), and IENFD of the hind paw (**E**). Statistical comparisons were made among the 5 groups by ANOVA and post hoc Tukey test. *** *p* < 0.001; HFD, Group #3 at 4 months compared to all other groups in all the parameters.

**Figure 6 ijms-26-00080-f006:**
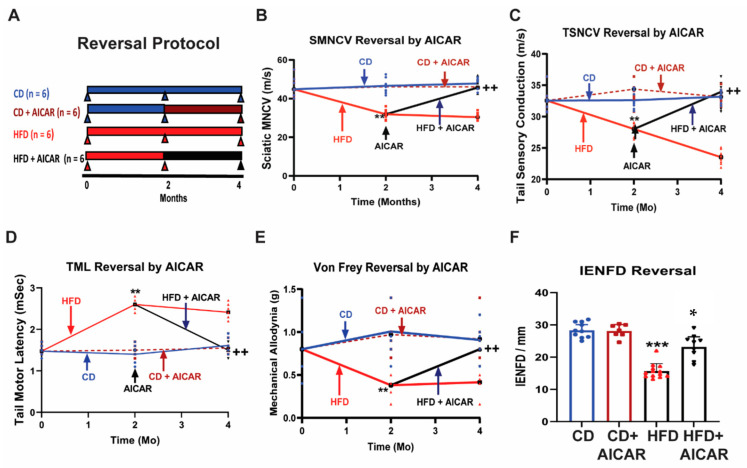
Two months of AICAR treatment reversed HFD-induced neuropathy in C57BL6 mice (*n* = 6/group). Three-month-old C57BL6 WT mice were fed with either a control diet (CD) or a high-fat diet (HFD). Baseline NCSs were completed at the beginning of the study. At 2 months, SMNCV, TML, TSNCV, and MA were measured in the mice fed a CD and the mice fed an HFD. After confirming that consumption of the HFD for 2 months induced development of peripheral neuropathy as observed by the changes in the NCSs and MA, AICAR was administered to the CD and HFD mice at a dose of 500 mg/kg for an additional 2 months. The vehicle was administered to CD and HFD mice. Nerve conduction studies were performed 2 months after administration of the AICAR or vehicle. The protocol is described in (**A**), the results of SMNCV (**B**), TSNCV (**C**), TML (**D**), MA using the Von Frey filament paw withdrawal threshold method (**E**), and intraepidermal fiber density (**F**) are shown. Statistical comparisons were made among the three groups with the ANOVA and post hoc Tukey test. * *p* < 0.05, ** *p* < 0.01, and *** *p* < 0.001; HFD mice at 2 months compared to 0-month-old HFD and CD mice in all parameters. ^++^ *p* < 0.01, HFD+AICAR mice at 4 months compared to HFD at 2 months in all parameters. The administration of AICAR reversed all the deficits of HFD-induced neuropathy. The administration of AICAR to non-diabetic mice had no significant effect. There was no statistically significant difference in CMAP or SNAP amplitudes compared between groups or compared at the start, during, and at the end of the treatment. The baseline (prior to starting any treatment) CMAP sciatic amplitude (mV) was 6.17 ± 0.78; CMAP tail amplitude (mV) was 3.050 ± 0.60, and SNAP tail amplitude (μV) was 51.92 ± 13.08. The CMAP sciatic amplitudes (mV) after 4 months of treatment were as follows: CD: 5.14 ± 0.99 (SEM); CD+AICAR: 5.70 ± 1.68; HFD: 8.31 ± 1.66; HFD+AICAR: 5.09 ± 0.76. The CMAP tail amplitudes (mV) after 4 months of treatment were as follows: CD: 1.24 ± 0.24; CD+AICAR: 2.16 ± 0.41; HFD: 2.96 ± 0.96; HFD+AICAR: 1.87 ± 0.70. The SNAP tail amplitudes (μV) after 4 months of treatment were as follows: CD: 102.48 ± 24.88; CD+AICAR: 126.77 ± 22.88; HFD: 56.21 ± 19.47; HFD+AICAR: 77.375 ± 19.46. There was no statistically significant difference in the CMAP or SNAP amplitudes during the reversal study, compared between groups or compared at the start and end of the treatment. This is as expected because of considerable variability in the CMAP and SNAP amplitudes [26,27,28].

**Figure 7 ijms-26-00080-f007:**
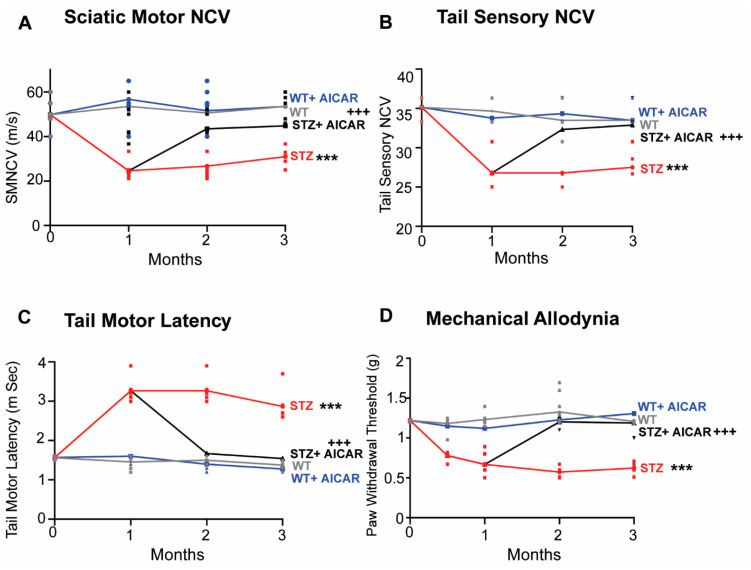
AICAR reverses STZ-induced neuropathy in C57BL6 mice (*n* = 6/group). Three-month-old C57BL6 WT and STZ-induced diabetic mice were purchased from Jackson Labs. The mice were fed with a control diet for a month. Measurement of NCSs and MA after a month showed that the STZ mice had developed neuropathy. Some of the STZ mice were then administered AICAR at a dose of 500 mg/kg for an additional 2 months. NCSs were performed at 5 and 6 months of age, namely after 2 and 3 months of STZ-induced diabetes or after 1 or 2 months of AICAR treatment. The results are shown as follows: SMNCV (**A**), TSNCV (**B**), TML (**C**), and MA (**D**) using the Von Frey filament paw withdrawal threshold method. Statistical comparisons were made among the three groups with the ANOVA and post hoc Tukey test. *** *p* < 0.001; STZ at experimental time 3 months compared to 1 month in all parameters. ^+++^ *p* < 0.001, STZ+AICAR at experimental time 3 months compared to STZ at 1 month in all parameters. Thus, administration of AICAR reversed all the peripheral neuropathy deficits of STZ mice.

**Table 1 ijms-26-00080-t001:** Metabolic end points in CD-fed and HFD-fed mice with and without AICAR treatment at 4 months.

Parameters	CD		HFD		Significance			
	(*n* = 6) 1	+AICAR (*n* = 6) 2	(*n* = 6) 3	+AICAR (*n* = 6) 4	1 vs. 2	1 vs. 3	2 vs. 4	3 vs. 4
Body Weight (g)	29 ± 3	28 ± 3	39 ± 5	29 ± 5	NS	<0.05	NS	<0.05
Plasma Glucose (mg/dL)	108 ± 20	111 ± 9	178 ± 9	138 ± 9	NS	<0.01	<0.05	<0.01
HbA1c%	5.4 ± 1	5 ± 0.6	6.8 ± 1	5.8 ± 0.9	NS	0.045	NS	<0.05
Insulin (ng/mL)	2.5 ± 0.2	2.3 ± 0.9	6.2 ± 1	3 ± 0.05	NS	<0.01	<0.05	<0.01
Total Cholesterol (mg/dL)	80 ± 4.2	78 ± 6.7	153 ± 9	133 ± 10	NS	<0.01	<0.01	<0.05
Triglycerides (mg/dL)	41 ± 6.1	39 ± 1.9	65 ± 5	46 ± 6	NS	<0.01	<0.05	<0.01
HDL Cholesterol (mg/dL)	70 ± 4	69 ± 5	143 ± 6	103 ± 5	NS	<0.01	<0.01	<0.05
HOMA-IR	5.0 ± 0.5	4.8 ± 0.4	10 ± 0.9	6.0 ± 0.6	NS	<0.001	<0.05	<0.001
NEFA (mM)	0.6 ± 0.06	0.6 ± 0.03	0.3 ± 0.03	0.2 ± 0.03	NS	<0.05	<0.01	<0.05
GTT-AUC (mg min/dL × 10^3^)	3.6 ± 0.46	3 ± 0.29	6.1 ± 1	4.1 ± 0.5	NS	<0.01	<0.05	<0.05

GTT-AUC = glucose tolerance test-area under the curve; HDL = high-density lipoprotein; HOMA-IR = HOMA-IR—an index used to determine if insulin resistance is present. NEFA = non-esterified fatty acids; NS = not significant.

## Data Availability

Data are available on request.

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
