# Peer review of "Administration of AICAR, an AMPK Activator, Prevents and Reverses Diabetic Polyneuropathy (DPN) by Regulating Mitophagy"

_ijms, 2024, doi:10.3390/ijms26010080_

Round 1

Reviewer 1 Report

Comments and Suggestions for Authors

REVIEW REPORT (ijms-3300257)

Title: Administration of AICAR, an AMPK Activator, Prevents and Reverses Diabetic Polyneuropathy (DPN) by Regulating Mitophagy

Abstract

Would it be possible for the authors to include numerical data of the main findings? This could be done if some of the text describing AMPK and mitophagy, which are mentioned repeatedly, were simplified.

Introduction

Please review the entire document to remove underlined sentences. Also, review the manuscript to make it more concise, without redundancies.

- The role of AICAR as an exercise mimetic is important, but what are the theoretical limitations of this approach? Are there known controversies or limitations in the literature that should be discussed? Can the authors adjust the writing to address these questions?

- You base the use of AICAR on previous studies, but how does this study differ and offer new information? Please improve this aspect.

Methods

- In general, the authors should better specify instruments and analysis techniques, such as the blotting conditions. This could be straightforward to do.

- The AICAR dosage used is high. Why was this dose chosen? Is there support in the literature for this dosage level in DPN models, and would it be safe in humans?

- The method of randomization and control is unclear. How did you ensure that randomization was done to minimize bias? This affects internal validity. If possible, improve the description to include more details about the randomization process.

- How were the key variables measured, especially AMPK phosphorylation and mitochondrial parameters?

Results

- No data on negative or neutral effects were mentioned. Were there any variables that did not respond to AICAR or discrepant results among mice?

Discussion

- I think it is risky to extrapolate to potential therapies in humans without clinical support. Please, you should moderate these statements in a way that respects the limitations of the model.

- I honestly think that the authors should address whether this dosage of AICAR is transposable to humans. Safety studies in murine models are insufficient for this purpose.

Conclusions

- The conclusion repeats the discussion and is redundant. The authors should make a small adjustment to adopt shorter sentences. Also, make sure to directly address the main findings and limitations. The conclusions point to the clinical potential of AICAR, but the limitations of the animal model need to be considered. Please reflect on this.

Author Response

Reviewer # 1

REVIEW REPORT (ijms-3300257)

Title: Administration of AICAR, an AMPK Activator, Prevents and Reverses Diabetic Polyneuropathy (DPN) by Regulating Mitophagy

Reviewer 1:

Abstract

Would it be possible for the authors to include numerical data of the main findings? This could be done if some of the text describing AMPK and mitophagy, which are mentioned repeatedly, were simplified.

Response: We have added numerical data on the main points in the abstract and text. We are limited by the word count to 200 words, so we have tightened the overall summary of the results. We have suggested the potential mechanism of protection by AICAR against DPN (line 24 to 32).

Introduction

Please review the entire document to remove underlined sentences. Also, review the manuscript to make it more concise, without redundancies.

Response: We have removed underlined sentences and replaced them with concise statements (line 44 to 45).

- The role of AICAR as an exercise mimetic is important, but what are the theoretical limitations of this approach? Are there known controversies or limitations in literature that should be discussed? Can the authors adjust the writing to address these questions?

Response: We have added the following section to the discussion:

AICAR (acadesine) has been used in human studies. AICAR has been used to treat surgical ischemia to reduce post perfusion myocardial infarction [1] and relapsing chronic leukemia [2]. In humans, doses up to 210 mg/kg IV are well tolerated [2]. At doses up to 100 mg/kg, only mild and transient side effects are re-ported equally in placebo and drug groups [3]. Thus, the dose used in this study in mice would fall well within the therapeutic human equivalent dose [4]. At doses greater than 200 mg/kg, adverse effects included hyperuricemia that occurred commonly but was not clinically significant and resolved with the administration of prophylactic allopurinol [2]. Other adverse events included transient anemia and/or thrombo-cytopenia (not clinically significant), renal impairment, and transient infusion-related hypotension (clinically significant) [2].  Furthermore, AICAR was well tolerated in more than 4000 cardiac patients [1, 3, 5-7]. Unfortunately, the energy promoting effects of AICAR have led to abuse in human athletes and animals involved in sports [8]. To place this is context, AICAR is a normal cellular intermediate in the human body [9] at concentrations necessary for human cellular activity, and is orally active [10] although bioavailability is poor [3]. Thus, at physiological concentrations, it has a non-toxic physiological concentration in the body but if abused at high, sustained dos-es would be expected to be toxic. However, there is interest in developing direct AMPK activator that may be safer. For example, DW14006, administered at a dose of 15 or 30 mg/kg by i.p. injection for 4 weeks has been shown to ameliorate DPN by regulating mitochondrial dysfunction, oxidative stress, and inflammation [11].

AICAR is prohibited in sport because of abuse in uncontrolled circumstances by athletes, and in particular cyclists. However, we do not know the doses used and most of the drug used was not controlled by the FDA so we know nothing about purity, contaminants etc.

In this manuscript, we wanted to establish proof of principle. Further research is being done to identify molecules that activate AMPK in a way that allows for the treatment of diseases in humans with minimal side effects, particularly in diabetes and heart disease. For example, A direct AMPK activator, DW14006, administered at a dose of 15 or 30 mg/kg by i.p. injection for 4 weeks has been shown to ameliorate DPN by regulating mitochondrial dysfunction, oxidative stress, and inflammation [11]

- You base the use of AICAR on previous studies, but how does this study differ and offer new information? Please improve this aspect.

Response: Increased physical activity has demonstrated positive effects in preventing and ameliorating brain disorders such as Alzheimer disease. Therefore, previous studies focused on the effect of AICAR on CNS. Enhancing the regeneration of axons is a reasonable therapeutic target for improving functional recovery after peripheral nerve injury. Several clinical studies have suggested exercise as a non-pharmacological approach to positively affect various aspects of peripheral neuropathy such as diabetic peripheral neuropathy (DPN). Unfortunately as explained in the manuscript, many patients with DPN cannot exercise and thus testing an exercise mimetic in DPN is novel and timely and there is no specific therapy for DPN. Relatively little is known about the effect of exercise mimetics such as AICAR on the regenerative effect in the peripheral nervous system.  Our study fulfills this missing information. 

Methods

- In general, the authors should better specify instruments and analysis techniques, such as the blotting conditions. This could be straightforward to do.

Response: Many of these methods are very length, further adding to the long Methods section, and have been extensively described in our previous publications. Our previous publications have provided detailed description of the instruments and analysis techniques and we have cited these in the references.

- The AICAR dosage used is high. Why was this dose chosen? Is there support in the literature for this dosage level in DPN models, and would it be safe in humans?

Response: In mice studies, AICAR dosages used by several investigators, safely with biological efficacy in rodents, vary from 250 mg/kg to 500 mg/kg. At the start of the study, we were aware of the effects of AICAR on diabetes studies but not on DPN. We wanted to test the proof of principle in this paper. Although we cannot exactly calculate the human equivalent dose, the human equivalent dose would be approximately 42 mg/kg for a 60 kg person. Humans tolerate up to 210 mg/kg at least with minimal adverse effects [2]. Thus 500 mg/kg in mice is well within the toxicity range for humans. Oral AICAR is biologically active but with poor absorption. Development of alternate AMPK activators that are useful at lower dose are being tested. For example, a direct AMPK activator, DW14006, administered at a dose of 15 or 30 mg/kg by i.p. injection for 4 weeks in mice has been shown to ameliorate DPN by regulating mitochondrial dysfunction, oxidative stress, and inflammation [11]

- The method of randomization and control is unclear. How did you ensure that randomization was done to minimize bias? This affects internal validity. If possible, improve the description to include more details about the randomization process.

Response: Randomization was performed based on a computer algorithm and all groups were compared at baseline to ensure that weight and electrophysiology was not statistically different between groups. To ensure that our results are robust, unbiased, and reproducible, the following was performed for all experiments: 1) random assignment of mice; 2) performance of behavioral studies by personnel blinded to the identity of the groups; 3) blind assessment of outcome measures; 4) specific exclusion criteria identified prior to initiation of all experiments (lines 441-445).

- How were the key variables measured, especially AMPK phosphorylation and mitochondrial parameters?

Response: The methods are described under materials and methods Section 4.3, 4.4 and 4.5. (A): Total AMPK and phosphorylated AMPK were quantified by measuring their levels of expression by Western Blots of proteins extracted from DRG neurons of mice fed with either CD or HFD. Equal amounts of DRG proteins (30 μg) were electrophoresed on SDS-PAGE gels and transferred to PVDF membranes. The membranes after blocking were incubated overnight at 4°C with different primary antibodies obtained from reliable sources: Cell Signaling: for total AMPKα Antibody #2532; for Phospho-AMPKα (Thr172) Antibody #2531 and control β-Actin (13E5) Rabbit mAb #4970. the membranes were incubated with a secondary antibody, peroxidase-conjugated goat anti-rabbit IgG (H+L). The membranes were developed with an advanced reagent (Bio-Rad, USA), and the protein bands were visualized with an automatic chemiluminescence apparatus (Bio-Rad, USA). The densities of the bands were determined with the Bio-Rad software. (B): Isolated mitochondria were diluted in cold 1x respiration buffer (230mM mannitol, 70mM sucrose, 0.02mM EDTA, 20mM Tris-HCl, 5mM K2HPO4 at pH 7.4, 37°C) and transferred to a Seahorse 24-well plate (10ug Mt in each well). Malate and pyruvate were used as substrates to assess Complex I-mediated respiration.  State 3 respiration was initiated by the addition of 40mM ADP.  Approximately 2 min later, state 3 respiration was terminated and state 4o respiration (resting) was initiated with addition of 20µM oligomycin, an inhibitor of the mitochondrial ATP synthase.  The maximal rate of uncoupled respiration was subsequently measured by the addition of 40µM carbonyl cyanide p-(trifluoromethoxy)phenylhydrazone (FCCP), which is a protonophore uncoupling molecule.  Rates are an average of 4–6 independent experiments for complex I respiration (lines 527-538).

Results

- No data on negative or neutral effects were mentioned. Were there any variables that did not respond to AICAR or discrepant results among mice?

Response: There seems to be no difference in the nerve conduction CMAP and SNAP amplitudes but this is likely related to a high coefficient of variation previously noted in these specific measurements and is a problem with performing electrophysiology in tiny animals. However, similar discrepancies are observed in humans. These results are included in the results section. Furthermore, the study only reports on use of AICAR up to 4 months and does not report on long term use of AICAR and how this is tolerated. Toxic effects of AICAR in ischemic condition are observed. This is more likely to be related to non-AMPK effects of AICAR on nucleotide synthesis and cell cycle arrest than on the activation of AMPK that is usually associated with increased oxidative metabolism. Therefore, AICAR is potentially useful in conditions where the ratio of AMP to ATP is low and therefore AICAR can stimulate AMPK to the max to reduce non-AMPK effects.

Discussion

- I think it is risky to extrapolate to potential therapies in humans without clinical support. Please, you should moderate these statements in a way that respects the limitations of the model.

- I honestly think that the authors should address whether this dosage of AICAR is transposable to humans. Safety studies in murine models are insufficient for this purpose.

Response: We agree and have moderated the conclusions and discussion. In mice studies, AICAR dosage used by several investigators vary from 250 mg/kg to 500 mg/kg. We have added a new paragraph addressing the use of AICAR in humans and the translations from mice to humans in section 3.5 lines 407-427.  Although we cannot exactly calculate the human equivalent dose, the human equivalent dose would be approximately 42 mg/kg for a 60 kg person. Humans tolerate up to 210 mg/kg AICAR at least. This is well within the toxicity range for humans.

Conclusions

- The conclusion repeats the discussion and is redundant. The authors should make a small adjustment to adopt shorter sentences. Also, make sure to directly address the main findings and limitations. The conclusions point to the clinical potential of AICAR, but the limitations of the animal model need to be considered. Please reflect on this.

Response: The conclusions were re-worded as follows:

In summary, the administration of AICAR increased phosphorylation of AMPK. Phosphorylated AMPK, in turn, activated proteins involved in the segregation of mitochondria by promoting mitochondrial fission. Administration of AICAR also influenced insulin sensitivity, glucose and lipid metabolism. These metabolic effects, in turn, protected and reversed DPN in both T1D and T2D mouse models. Drugs targeting AMPK phosphorylation, such as AICAR, may function as exercise mimetics. Hu-mans with DPN are often disabled and cannot exercise adequately. Thus, exercise mimetics could play a crucial role as therapy in DPN, but this would require further clinical trials in humans to assess this. AICARs direct AMPK activation and transport via ENT1 make it an intriguing candidate for further investigation to develop iso-form-specific AMPK activators in the nervous system. The neuroprotective properties of AMPK activators could be utilized in DPN but further studies are needed in humans (lines 556-568).

  1. Mangano, D. T.; Miao, Y.; Tudor, I. C.; Dietzel, C.; Investigators of the Multicenter Study of Perioperative Ischemia Research, G.; Ischemia, R.; Education, F., Post-reperfusion myocardial infarction: long-term survival improvement using adenosine regulation with acadesine. J Am Coll Cardiol 2006, 48, (1), 206-14.
  2. Van Den Neste, E.; Cazin, B.; Janssens, A.; Gonzalez-Barca, E.; Terol, M. J.; Levy, V.; Perez de Oteyza, J.; Zachee, P.; Saunders, A.; de Frias, M.; Campas, C., Acadesine for patients with relapsed/refractory chronic lymphocytic leukemia (CLL): a multicenter phase I/II study. Cancer Chemother Pharmacol 2013, 71, (3), 581-91.
  3. Dixon, R.; Gourzis, J.; McDermott, D.; Fujitaki, J.; Dewland, P.; Gruber, H., AICA-riboside: safety, tolerance, and pharmacokinetics of a novel adenosine-regulating agent. J Clin Pharmacol 1991, 31, (4), 342-7.
  4. Nair, A. B.; Jacob, S., A simple practice guide for dose conversion between animals and human. J Basic Clin Pharm 2016, 7, (2), 27-31.
  5. Mangano, D. T., Effects of acadesine on myocardial infarction, stroke, and death following surgery. A meta-analysis of the 5 international randomized trials. The Multicenter Study of Perioperative Ischemia (McSPI) Research Group. JAMA 1997, 277, (4), 325-32.
  6. Drew, B. G.; Kingwell, B. A., Acadesine, an adenosine-regulating agent with the potential for widespread indications. Expert Opin Pharmacother 2008, 9, (12), 2137-44.
  7. Mahaffey, K. W.; Puma, J. A.; Barbagelata, N. A.; DiCarli, M. F.; Leesar, M. A.; Browne, K. F.; Eisenberg, P. R.; Bolli, R.; Casas, A. C.; Molina-Viamonte, V.; Orlandi, C.; Blevins, R.; Gibbons, R. J.; Califf, R. M.; Granger, C. B., Adenosine as an adjunct to thrombolytic therapy for acute myocardial infarction: results of a multicenter, randomized, placebo-controlled trial: the Acute Myocardial Infarction STudy of ADenosine (AMISTAD) trial. J Am Coll Cardiol 1999, 34, (6), 1711-20.
  8. Pokrywka, A.; Cholbinski, P.; Kaliszewski, P.; Kowalczyk, K.; Konczak, D.; Zembron-Lacny, A., Metabolic modulators of the exercise response: doping control analysis of an agonist of the peroxisome proliferator-activated receptor delta (GW501516) and 5-aminoimidazole-4-carboxamide ribonucleotide (AICAR). J Physiol Pharmacol 2014, 65, (4), 469-76.
  9. Visnjic, D.; Lalic, H.; Dembitz, V.; Tomic, B.; Smoljo, T., AICAr, a Widely Used AMPK Activator with Important AMPK-Independent Effects: A Systematic Review. Cells 2021, 10, (5).
  10. Narkar, V. A.; Downes, M.; Yu, R. T.; Embler, E.; Wang, Y. X.; Banayo, E.; Mihaylova, M. M.; Nelson, M. C.; Zou, Y.; Juguilon, H.; Kang, H.; Shaw, R. J.; Evans, R. M., AMPK and PPARdelta agonists are exercise mimetics. Cell 2008, 134, (3), 405-415.
  11. Xu, X.; Wang, W.; Wang, Z.; Lv, J.; Xu, X.; Xu, J.; Yang, J.; Zhu, X.; Lu, Y.; Duan, W.; Huang, X.; Wang, J.; Zhou, J.; Shen, X., DW14006 as a Direct AMPKalpha Activator Ameliorates Diabetic Peripheral Neuropathy in Mice. Diabetes 2020, 69, (9), 1974-1988.
  12. Chandrasekaran, K.; Salimian, M.; Konduru, S. R.; Choi, J.; Kumar, P.; Long, A.; Klimova, N.; Ho, C. Y.; Kristian, T.; Russell, J. W., Overexpression of Sirtuin 1 protein in neurons prevents and reverses experimental diabetic neuropathy. Brain 2019, 142, (12), 3737-3752.
  13. Chandrasekaran, K.; Najimi, N.; Sagi, A. R.; Yarlagadda, S.; Salimian, M.; Arvas, M. I.; Hedayat, A. F.; Kevas, Y.; Kadakia, A.; Russell, J. W., NAD(+) Precursors Repair Mitochondrial Function in Diabetes and Prevent Experimental Diabetic Neuropathy. Int J Mol Sci 2022, 23, (9).
  14. Chandrasekaran, K.; Najimi, N.; Sagi, A. R.; Yarlagadda, S.; Salimian, M.; Arvas, M. I.; Hedayat, A. F.; Kevas, Y.; Kadakia, A.; Kristian, T.; Russell, J. W., NAD(+) Precursors Reverse Experimental Diabetic Neuropathy in Mice. Int J Mol Sci 2024, 25, (2).
  15. Chaudhry, V.; Cornblath, D. R.; Mellits, E. D.; Avila, O.; Freimer, M. L.; Glass, J. D.; Reim, J.; Ronnett, G. V.; Quaskey, S. A.; Kuncl, R. W., Inter- and intra-examiner reliability of nerve conduction measurements in normal subjects. Ann. Neurol 1991, 30, (6), 841-843.
  16. Biessels, G. J.; Bril, V.; Calcutt, N. A.; Cameron, N. E.; Cotter, M. A.; Dobrowsky, R.; Feldman, E. L.; Fernyhough, P.; Jakobsen, J.; Malik, R. A.; Mizisin, A. P.; Oates, P. J.; Obrosova, I. G.; Pop-Busui, R.; Russell, J. W.; Sima, A. A.; Stevens, M. J.; Schmidt, R. E.; Tesfaye, S.; Veves, A.; Vinik, A. I.; Wright, D. E.; Yagihashi, S.; Yorek, M. A.; Ziegler, D.; Zochodne, D. W., Phenotyping animal models of diabetic neuropathy: a consensus statement of the diabetic neuropathy study group of the EASD (Neurodiab). J Peripher. Nerv. Syst 2014, 19, (2), 77-87.
  17. O'Brien, J.; Niehaus, P.; Chang, K.; Remark, J.; Barrett, J.; Dasgupta, A.; Adenegan, M.; Salimian, M.; Kevas, Y.; Chandrasekaran, K.; Kristian, T.; Chellappan, R.; Rubin, S.; Kiemen, A.; Lu, C. P.; Russell, J. W.; Ho, C. Y., Skin keratinocyte-derived SIRT1 and BDNF modulate mechanical allodynia in mouse models of diabetic neuropathy. bioRxiv 2024.
  18. O'Brien, P. D.; Guo, K.; Eid, S. A.; Rumora, A. E.; Hinder, L. M.; Hayes, J. M.; Mendelson, F. E.; Hur, J.; Feldman, E. L., Integrated lipidomic and transcriptomic analyses identify altered nerve triglycerides in mouse models of prediabetes and type 2 diabetes. Dis Model Mech 2020, 13, (2).

Reviewer 2 Report

Comments and Suggestions for Authors

In this paper, the authors investigated the impact of an exercise mimetic, 5-aminoimidazole-4-carboxamide ribonucleotide (AICAR), on Diabetic neuropathy in mouse models of both T2D and T1D. AICAR treatment improved some alterations indicative of sensory neuropathy in both animal models. They further investigated some involved mechanisms in the T2D mice, finding that AICAR activates AMPK, that mediates phosphorylation and activation of proteins involved in mitochondrial quality and mitochondrial respiration. In T2D mice the improvement was also mediated by effects on insulin sensitivity and metabolism.

The study was well designed, and the methodology used is adequate, although some more information appears needed.

Comments:

-       It is not explained in Material and Methods, subsection 4.1 how were the control and T2D mice divided in groups and treated, as it is in 4.2 for the STZ-induced diabetic mice.

-       The protocol for inducing diabetes with streptozotocin should be indicated.

-       Which is the rationale for the dose of AICAR administered? In previous studies (Narkar et al 2009) the compound was given at 500mg/kg/day for 4 weeks, showing an effect on endurance and gene expression.

-       How can be explained that AICAR administration increased the phosphorylation of AMPK twice as much in HFD mice than in Control mice? Any particular interaction?

-       Comment on the reduction by AICAR of the levels of HDL-cholesterol in HFD mice (Table 1). Did you measure LDL-cholesterol also?

-       The evaluation of neuropathy was only performed at the end of the study, i.e. 4 months after starting treatment. It would have been better to perform functional tests at monthly intervals, or at least at 2 and 4 months in the same animals, to assess the evolution of the nerve involvement.

-       In the reversal study, values of NCV at 2 months in the HFD mice are comparatively lower than in the corresponding group of the prevention study at 4 months. This emphasizes the need for repeated testing (at least at 2 and 4 months) in all the mouse groups. In addition, how can be explained the fast decline in NCV at the first 2 months and the stabilization later?

-       Results of the amplitude of the SCNAP and the MCNAP must be reported for all the experimental groups. These values are the most relevant in diabetic neuropathy, since they correlate with the number of myelinated fibers degenerating or non-functioning. On the other hand, NCV is related with myelin involvement, but it is also affected by metabolic derangements.

-       Since there are evidence that diabetes affects largely myelinated nerve fibers, the assessment of Meissner’s corpuscles in the foot-pad samples stained with PGP9.5 can provide valuable information. Please add this information.

-       Which were the values for glycemia, GTT, and lipids in the STZ-diabetic and control mice?

-       The practically complete prevention or reversal of the neuropathic alterations by AICAR treatment does not correspond with the partial improvement of glycemia (as in Table 1). Can you comment on this discrepancy?

-       Previous studies reported that activation of AMPK by 5-aminoimidazole-4-carboxamide ribonucleoside exacerbated damage after stroke, indicating deleterious effects on neural insults. Does AICAR treatment induce neuronal damage or death in DRG under diabetic conditions?

-       The conclusions should be reworded since there are not enough evidence of improvement of neuropathy. Histological evaluation of nerve fibers in distal nerves, and amplitudes of the CNAPs will help to distinguish metabolic vs protective effects.

-       There are some expressions that need correction. For example, in p 12 line 407 “DRG of mouse brain”, in p 9, line 261 “sensory motor nerve conduction velocity (SMNCV)”

Author Response

Reviewer # 2

Open Review

Comments and Suggestions for Authors

In this paper, the authors investigated the impact of an exercise mimetic, 5-aminoimidazole-4-carboxamide ribonucleotide (AICAR), on Diabetic neuropathy in mouse models of both T2D and T1D. AICAR treatment improved some alterations indicative of sensory neuropathy in both animal models. They further investigated some involved mechanisms in the T2D mice, finding that AICAR activates AMPK, that mediates phosphorylation and activation of proteins involved in mitochondrial quality and mitochondrial respiration. In T2D mice the improvement was also mediated by effects on insulin sensitivity and metabolism.

The study was well designed, and the methodology used is adequate, although some more information appears needed.

Comments:

-       It is not explained in Material and Methods, subsection 4.1 how were the control and T2D mice divided in groups and treated, as it is in 4.2 for the STZ-induced diabetic mice.

Response: We have added further information on the methods used in the HFD mice in the Methods section 4.1: Randomization was performed based on a computer algorithm and all groups were compared at baseline to ensure that weight and electrophysiology was not statistically different between groups. To ensure that our results are robust, unbiased, and reproducible, the following was performed for all experiments: 1) random assignment of mice; 2) performance of behavioral studies by personnel blinded to the identity of the groups; 3) blind assessment of outcome measures; 4) specific exclusion criteria identified prior to initiation of all experiments (lines 441-445).

-       The protocol for inducing diabetes with streptozotocin should be indicated.

Response: We have added these changes to the methods. Three-month-old, male, C57BL6 WT and Streptozotocin- (STZ-) induced diabetic mice were purchased from Jackson Labs. Six-week-old males were identified, weighed, and a baseline non-fasted glucose measurement was taken with a OneTouch Ultra2 or Ultra/Mini glucometer. Mice received daily IP injections of 50 mg STZ/kg body weight (C57BL/6J) for five consecutive days; age-matched controls received buffer injections.  Mice were housed in disposable cages with absorbent bedding and ad libitum access to food and water while they metabolized STZ for at least 48 hours after the final injection. After transfer to regular cages, mice were observed daily then weighed and glucose-tested 7-14 days after the final injection. In addition, mice that appear overtly healthy without significant changes in body weight were used. (lines 468-478).

-       Which is the rationale for the dose of AICAR administered? In previous studies (Narkar et al 2009) the compound was given at 500mg/kg/day for 4 weeks, showing an effect on endurance and gene expression.

Response: mice studies, AICAR dosages used by several investigators, safely with biological efficacy in rodents, vary from 250 mg/kg to 500 mg/kg. At the start of the study, we were aware of the effects of AICAR on diabetes studies but not on DPN. We wanted to test the proof of principle in this paper. Although we cannot exactly calculate the human equivalent dose, the human equivalent dose would be approximately 42 mg/kg for a 60 kg person. Humans tolerate up to 210 mg/kg at least with minimal adverse effects [2]. Thus 500 mg/kg in mice is well within the toxicity range for humans. Oral AICAR is biologically active but with poor absorption. We have added a detailed paragraph in section 3.5 explaining the human and equivalency dosing of AICAR.

-       How can be explained that AICAR administration increased the phosphorylation of AMPK twice as much in HFD mice than in Control mice? Any particular interaction?

Response: Yes, the overstimulation of AMPK by AICAR in HFD-fed mice is likely due to nucleoside/nucleotide interactions. AICAR is phosphorylated to ZMP (AICAR monophosphate, aka AICA-ribotide) by adenosine kinase. ZMP, an AMP analog and an endogenous precursor of inosine monophosphate (IMP) in the de novo pathway of purine synthesis, binds to the γ-subunit and activates AMPK. Likewise, the entry of AICAR into cells is also regulated by nucleotide transporter ENT1. Therefore, altered nucleoside/nucleotide levels in HFD-mice could regulate and contribute to the overstimulation of phosphorylation of AMPK by AICAR in HFD mice. 

-       Comment on the reduction by AICAR of the levels of HDL-cholesterol in HFD mice (Table 1). Did you measure LDL-cholesterol also?

Response: Long-term administration of AICAR is shown to reduce plasma triglyceride levels, non-esterified free fatty acids and increase HDL-cholesterol, the decrease seen in HFD+AICAR may possibly relate to the duration of treatment. We did not measure LDL-cholesterol specifically (lines 172-185).

-       The evaluation of neuropathy was only performed at the end of the study, i.e. 4 months after starting treatment. It would have been better to perform functional tests at monthly intervals, or at least at 2 and 4 months in the same animals, to assess the evolution of the nerve involvement.

Response: In the prevention study, mice were randomly assigned into 4 groups and the effect of AICAR on neuropathy was only performed at 4 months. There is likely to be individual variations and therefore the effects may appear much smaller. Change with neuropathy over time is addressed in the longitudinal reversal study.

-       In the reversal study, values of NCV at 2 months in the HFD mice are comparatively lower than in the corresponding group of the prevention study at 4 months. This emphasizes the need for repeated testing (at least at 2 and 4 months) in all the mouse groups. In addition, how can be explained the fast decline in NCV at the first 2 months and the stabilization later?

Response: In HFD-fed mice, there were individual variations in nerve conduction parameters. The mice were evaluated at 2 months. Mice which showed a more rapid change at 2 months in nerve conduction parameters were then tested for the reversal with AICAR.  Therefore, the changes in nerve conduction parameters are much more severe than in the prevention study. In all the studies we have performed in HFD mice, there is an initial rapid initial decline in the sciatic motor conduction velocities or tail sensory conduction velocities or tail motor latencies followed by a more gradual decline [12-14]. In the current study, the mean values for the NCS are fairly close at 4 months in the prevention and reversal study, except for the TSNCV for the HFD group. at and are not statistically significant between the prevention and reversal studies.

-       Results of the amplitude of the SCNAP and the MCNAP must be reported for all the experimental groups. These values are the most relevant in diabetic neuropathy, since they correlate with the number of myelinated fibers degenerating or non-functioning. On the other hand, NCV is related with myelin involvement, but it is also affected by metabolic derangements.

Response:

This is an important concern. We have reported these values in the results. Compound muscle action potentials (CMAPs) were measured from the motor stimulation as the recording electrodes were placed in muscle distal to the site of stimulation. Sensory nerve action potentials (SNAPs) are near nerve potentials with stimulation and recording close to the nerve with needle electrodes. Unfortunately, the CMAPs and SNAPs show significant variability in amplitude and area and this affects their use in determining changes in both mouse and human nerve in type 2 diabetes mellitus. This variability and lack of reproducibility has been well documented and thus, it is recommended to use nerve conduction velocities or latencies where possible [15, 16]. We looked at all the waveforms and did not find any evidence consistent with demyelination or significant degeneration of the myelin sheath. The lack of reproducibility and reliability of sequential CMAP and SNAP amplitudes was addressed by an expert group. The NIDDK, JDRF, and the Diabetic Neuropathy Study Group of EASD sponsored a meeting to explore the current status of animal models of diabetic peripheral neuropathy[16]. A neuropathy phenotype in rodents was defined as the presence of statistically different values between diabetic and control animals in 2 of 3 assessments (nocifensive behavior, nerve conduction velocities or nerve structure). All of these measurements were used in the current study.

-       Since there are evidence that diabetes affects largely myelinated nerve fibers, the assessment of Meissner’s corpuscles in the foot-pad samples stained with PGP9.5 can provide valuable information. Please add this information.

Response: Although, in this study, we did not assess the role of Meissner’s corpuscles in the foot-pad samples stained with PGP9.5. We have published results showing the role of Meissner’s corpuscles in two mouse models of HFD and HFD+STZ. In the skin biopsy of these mice there was a profound loss of Meissner corpuscles (sensory mechanoreceptors which normally signal light touch and vibration) as well as degeneration or retraction of the Aβ sensory axons which innervate them [17]

-       Which were the values for glycemia, GTT, and lipids in the STZ-diabetic and control mice?

Response:

In order not to be repetitive we did not include blood values for STZ-diabetic mice in this manuscript, but results are published in our previous publication Table 2 in
NAD+ Precursors Repair Mitochondrial Function in Diabetes and Prevent Experimental Diabetic Neuropathy. Chandrasekaran K, Najimi N, Sagi AR, Yarlagadda S, Salimian M, Arvas MI, Hedayat AF, Kevas Y, Kadakia A, Russell JW.Int J Mol Sci. 2022 Apr 28;23(9):4887. doi: 10.3390/ijms23094887.PMID: 35563288.

Parameters

Significance

Group #

Non-Diabetic

(n = 6)

1

STZ

(n=6)

2

1 vs 2

Body Weight (g)

30 ± 2

23 ± 3

<0.01

Plasma Glucose (mg/dL)

105 ± 12

405 ± 32

<0.001

HbA1c%

5.4 ± 1

15.7 ± 3.1

<0.001

Insulin (µg/L)

0.8 ± 0.2

0.2 ± 0.05

<0.001

Total Cholesterol (mg/dL)

78 ± 13

173 ± 42

<0.01

Triglycerides (mg/dL)

44 ± 8

93 ± 13

<0.001

HDL (mg/dL)

79 ± 4

108 ± 21

<0.01

LDL (mg/dL)

43 ± 5

83 ± 5

<0.01

NEFA (mM)

2.5 ± 0.8

6.4 ± 1.4

<0.001

-       The practically complete prevention or reversal of the neuropathic alterations by AICAR treatment does not correspond with the partial improvement of glycemia (as in Table 1). Can you comment on this discrepancy?

Response: Recent research, prompted by clinical and animal model studies, has shown that in T2D, in addition to the classical pathways activated by hyperglycemia, dyslipidemia plays an important role in the development of DPN. Overall glucose control may only have a moderate impact on DPN in T2D. Therefore, we suggest that abnormal nerve-lipid signaling may be an important factor in T2D DPN. Administration of AICAR affects this lipid signaling as evidenced by lowered triglycerides, HDL-cholesterol, HOMA-IR and NEFA (Table 1). In other studies, AICAR was found to be able to inhibit the synthesis of fatty acids and enhance fatty acid oxidation. Altered nerve triglycerides is observed in mouse models of prediabetes and T2D (Integrated lipidomic and transcriptomic analyses identify altered nerve triglycerides in mouse models of prediabetes and type 2 diabetes [18]. This may provide explanation as to how improved lipid signaling and partial improvement in glycemia by AICAR prevents and reverses DPN. Furthermore, AICAR likely acts directly in the nerve to reduce the hyperglycemic axonal degeneration and this would occur independently of any change in the serum glucose concentration.

-       Previous studies reported that the activation of AMPK by 5-aminoimidazole-4-carboxamide ribonucleoside exacerbated damage after stroke, indicating deleterious effects on neural insults. Does AICAR treatment induce neuronal damage or death in DRG under diabetic conditions?

Response: AICAR treatment did not induce neuronal damage or death in DRG under diabetic conditions. This contrasts with a previous study (Pharmacological Inhibition of AMP-activated Protein Kinase Provides Neuroprotection in Stroke, The Journal of Biological Chemistry, Vol. 280, No. 21, Issue of May 27, pp. 20493–20502, 2005), showing increased neuronal death in stroke following AICAR treatment. In that study, AMPK was shown to be highly expressed in cortical and hippocampal neurons under both normal and ischemic conditions. AMPK activity, as assessed by the phosphorylation of AMPK, namely pAMPK was increased following both middle cerebral artery occlusion and oxygen-glucose deprivation. Whereas in our study, the level of pAMPK is decreased in diabetic conditions and addition of AICAR increased its level. This contrast between these two conditions is likely due to the differences in the ratio of AMP to ATP, in ischemic condition the ratio is likely to be high due to increased consumption of ATP possibly to maintain neuronal membrane potential and therefore pAMPK is already activated by AMP-mediated phosphorylation. In contrast, in diabetic conditions, the ratio of AMP to ATP is likely to be low, and that is why addition of AICAR activates AMPK to the max and therefore the level of pAMPK is high. This suggests a hypothesis that the toxic effects of AICAR in ischemic condition are more likely to be related to non-AMPK effects of AICAR on nucleotide synthesis and cell cycle arrest than on the activation of AMPK that is usually associated with increased oxidative metabolism.

  • The conclusions should be reworded since there are not enough evidence of improvement of neuropathy. Histological evaluation of nerve fibers in distal nerves, and amplitudes of the CNAPs will help to distinguish metabolic vs protective effects.

Response: We respectfully disagree. The current study shows that the intraepidermal nerve fiber density, which is a clear measure of unmyelinated fiber density and neuropathy in DSP is clearly reduced with a HFD and clearly shows that the decline in IENFD is reduced when the animals are then treated with AICAR as part of the reversal study. Furthermore, NCS along with measures of nociception have been widely used to measure worsening or improvement in diabetic neuropathy. We have however reworded the conclusions:

  In summary, the administration of AICAR increased phosphorylation of AMPK. Phosphorylated AMPK, in turn, activated proteins involved in the segregation of mitochondria by promoting mitochondrial fission. Administration of AICAR also influenced insulin sensitivity, glucose and lipid metabolism. These metabolic effects, in turn, protected and reversed DPN in both T1D and T2D mouse models. Drugs targeting AMPK phosphorylation, such as AICAR, may function as exercise mimetics. Humans with DPN are often disabled and cannot exercise adequately. Thus, exercise mimetics could play a crucial role as therapy in DPN, but this would require further clinical trials in humans to assess this. AICARs direct AMPK activation and transport via ENT1 make it an intriguing candidate for further investigation to develop iso-form-specific AMPK activators in the nervous system. The neuroprotective properties of AMPK activators could be utilized in DPN but further studies are needed in humans (Lines 557-568)

  • There are some expressions that need correction. For example, in p 12 line 407 “DRG of mouse brain”, in p 9, line 261 “sensory motor nerve conduction velocity (SMNCV)”

Response: Thanks for pointing this out, we have made these corrections.

  1. Mangano, D. T.; Miao, Y.; Tudor, I. C.; Dietzel, C.; Investigators of the Multicenter Study of Perioperative Ischemia Research, G.; Ischemia, R.; Education, F., Post-reperfusion myocardial infarction: long-term survival improvement using adenosine regulation with acadesine. J Am Coll Cardiol 2006, 48, (1), 206-14.
  2. Van Den Neste, E.; Cazin, B.; Janssens, A.; Gonzalez-Barca, E.; Terol, M. J.; Levy, V.; Perez de Oteyza, J.; Zachee, P.; Saunders, A.; de Frias, M.; Campas, C., Acadesine for patients with relapsed/refractory chronic lymphocytic leukemia (CLL): a multicenter phase I/II study. Cancer Chemother Pharmacol 2013, 71, (3), 581-91.
  3. Dixon, R.; Gourzis, J.; McDermott, D.; Fujitaki, J.; Dewland, P.; Gruber, H., AICA-riboside: safety, tolerance, and pharmacokinetics of a novel adenosine-regulating agent. J Clin Pharmacol 1991, 31, (4), 342-7.
  4. Nair, A. B.; Jacob, S., A simple practice guide for dose conversion between animals and human. J Basic Clin Pharm 2016, 7, (2), 27-31.
  5. Mangano, D. T., Effects of acadesine on myocardial infarction, stroke, and death following surgery. A meta-analysis of the 5 international randomized trials. The Multicenter Study of Perioperative Ischemia (McSPI) Research Group. JAMA 1997, 277, (4), 325-32.
  6. Drew, B. G.; Kingwell, B. A., Acadesine, an adenosine-regulating agent with the potential for widespread indications. Expert Opin Pharmacother 2008, 9, (12), 2137-44.
  7. Mahaffey, K. W.; Puma, J. A.; Barbagelata, N. A.; DiCarli, M. F.; Leesar, M. A.; Browne, K. F.; Eisenberg, P. R.; Bolli, R.; Casas, A. C.; Molina-Viamonte, V.; Orlandi, C.; Blevins, R.; Gibbons, R. J.; Califf, R. M.; Granger, C. B., Adenosine as an adjunct to thrombolytic therapy for acute myocardial infarction: results of a multicenter, randomized, placebo-controlled trial: the Acute Myocardial Infarction STudy of ADenosine (AMISTAD) trial. J Am Coll Cardiol 1999, 34, (6), 1711-20.
  8. Pokrywka, A.; Cholbinski, P.; Kaliszewski, P.; Kowalczyk, K.; Konczak, D.; Zembron-Lacny, A., Metabolic modulators of the exercise response: doping control analysis of an agonist of the peroxisome proliferator-activated receptor delta (GW501516) and 5-aminoimidazole-4-carboxamide ribonucleotide (AICAR). J Physiol Pharmacol 2014, 65, (4), 469-76.
  9. Visnjic, D.; Lalic, H.; Dembitz, V.; Tomic, B.; Smoljo, T., AICAr, a Widely Used AMPK Activator with Important AMPK-Independent Effects: A Systematic Review. Cells 2021, 10, (5).
  10. Narkar, V. A.; Downes, M.; Yu, R. T.; Embler, E.; Wang, Y. X.; Banayo, E.; Mihaylova, M. M.; Nelson, M. C.; Zou, Y.; Juguilon, H.; Kang, H.; Shaw, R. J.; Evans, R. M., AMPK and PPARdelta agonists are exercise mimetics. Cell 2008, 134, (3), 405-415.
  11. Xu, X.; Wang, W.; Wang, Z.; Lv, J.; Xu, X.; Xu, J.; Yang, J.; Zhu, X.; Lu, Y.; Duan, W.; Huang, X.; Wang, J.; Zhou, J.; Shen, X., DW14006 as a Direct AMPKalpha Activator Ameliorates Diabetic Peripheral Neuropathy in Mice. Diabetes 2020, 69, (9), 1974-1988.
  12. Chandrasekaran, K.; Salimian, M.; Konduru, S. R.; Choi, J.; Kumar, P.; Long, A.; Klimova, N.; Ho, C. Y.; Kristian, T.; Russell, J. W., Overexpression of Sirtuin 1 protein in neurons prevents and reverses experimental diabetic neuropathy. Brain 2019, 142, (12), 3737-3752.
  13. Chandrasekaran, K.; Najimi, N.; Sagi, A. R.; Yarlagadda, S.; Salimian, M.; Arvas, M. I.; Hedayat, A. F.; Kevas, Y.; Kadakia, A.; Russell, J. W., NAD(+) Precursors Repair Mitochondrial Function in Diabetes and Prevent Experimental Diabetic Neuropathy. Int J Mol Sci 2022, 23, (9).
  14. Chandrasekaran, K.; Najimi, N.; Sagi, A. R.; Yarlagadda, S.; Salimian, M.; Arvas, M. I.; Hedayat, A. F.; Kevas, Y.; Kadakia, A.; Kristian, T.; Russell, J. W., NAD(+) Precursors Reverse Experimental Diabetic Neuropathy in Mice. Int J Mol Sci 2024, 25, (2).
  15. Chaudhry, V.; Cornblath, D. R.; Mellits, E. D.; Avila, O.; Freimer, M. L.; Glass, J. D.; Reim, J.; Ronnett, G. V.; Quaskey, S. A.; Kuncl, R. W., Inter- and intra-examiner reliability of nerve conduction measurements in normal subjects. Ann. Neurol 1991, 30, (6), 841-843.
  16. Biessels, G. J.; Bril, V.; Calcutt, N. A.; Cameron, N. E.; Cotter, M. A.; Dobrowsky, R.; Feldman, E. L.; Fernyhough, P.; Jakobsen, J.; Malik, R. A.; Mizisin, A. P.; Oates, P. J.; Obrosova, I. G.; Pop-Busui, R.; Russell, J. W.; Sima, A. A.; Stevens, M. J.; Schmidt, R. E.; Tesfaye, S.; Veves, A.; Vinik, A. I.; Wright, D. E.; Yagihashi, S.; Yorek, M. A.; Ziegler, D.; Zochodne, D. W., Phenotyping animal models of diabetic neuropathy: a consensus statement of the diabetic neuropathy study group of the EASD (Neurodiab). J Peripher. Nerv. Syst 2014, 19, (2), 77-87.
  17. O'Brien, J.; Niehaus, P.; Chang, K.; Remark, J.; Barrett, J.; Dasgupta, A.; Adenegan, M.; Salimian, M.; Kevas, Y.; Chandrasekaran, K.; Kristian, T.; Chellappan, R.; Rubin, S.; Kiemen, A.; Lu, C. P.; Russell, J. W.; Ho, C. Y., Skin keratinocyte-derived SIRT1 and BDNF modulate mechanical allodynia in mouse models of diabetic neuropathy. bioRxiv 2024.
  18. O'Brien, P. D.; Guo, K.; Eid, S. A.; Rumora, A. E.; Hinder, L. M.; Hayes, J. M.; Mendelson, F. E.; Hur, J.; Feldman, E. L., Integrated lipidomic and transcriptomic analyses identify altered nerve triglycerides in mouse models of prediabetes and type 2 diabetes. Dis Model Mech 2020, 13, (2).

Reviewer 3 Report

Comments and Suggestions for Authors

Dear authors,

Your research is very interesting and topical from my point of view. I would like to make a few small observations:

I would suggest the authors to introduce a figure related to the mechanism of action of AMP-activated protein kinases (the first paragraph on page 2), for an easier follow-up of the processes. Also, on the same figure, the action targets of the various drugs that shape AMPK activity could be marked (paragraph 2, page 2).

Is there no possibility of correlating the effects obtained following the use of substances that mimic physical exercises with physical exercises as such (that is, mice that are subjected to a daily physical effort)? In my opinion, I think that in terms of efficiency, such a comparison would be useful.

My recommendation for this research is to publish with minor revisions.

Author Response

Reviewer 3

Your research is very interesting and topical from my point of view. I would like to make a few small observations:

  • I would suggest the authors to introduce a figure related to the mechanism of action of AMP-activated protein kinases (the first paragraph on page 2), for an easier follow-up of the processes. Also, on the same figure, the action targets of the various drugs that shape AMPK activity could be marked (paragraph 2, page 2).

Response: Hypothesis figure added (lines 76 to 89).

Figure 1. Exercise consumes a large amount of ATP, elevating AMP to ATP ratio. Increased AMP binds to the enzyme AMPK and induces the phosphorylation of AMPK. AICAR directly phosphorylates AMPK. (1) Phosphorylated AMPK activates SIRT1/PGC1-α to the catabolic process of exercise (increased OXPHOS), resulting in decreased body weight, reduced triglycerides, improved HOMA-IR Index and lower NEFA levels, (2). Phosphorylated AMPK led to phosphorylation of mitochondrial fission factor (MFF), recruitment of dynamin-like protein DRP1 to mitochondria, and activation of ULK1, an upstream kinase in autophagy and mitophagy. Mitochondrial fission allows damaged mitochondria to be selectively degraded through mitophagy pathways. (3) Phosphorylated AMPK activates PGC-1α form co-transcriptional complexes that initiate the overexpression of target genes to promote myogenesis, neurogenesis, and mitochondrial bioenergetics. The main points of this paper are boxed in red.

  • Is there no possibility of correlating the effects obtained following the use of substances that mimic physical exercises with physical exercises as such (that is, mice that are subjected to a daily physical effort)? In my opinion, I think that in terms of efficiency, such a comparison would be useful.

Response: This is an excellent suggestion; we have not done those experiments but will do additional studies in the future to compare exercise with exercise mimetics to correlate their efficiency. We have added this section in the discussion: ” We did not address the combined effect of exercise and AICAR, because the premise is that many patients with DPN cannot exercise because of significant neuro-pathic pain and poor balance and therefore need an alternative to exercise to reduce the effect of diabetic neuropathy. However, this has been tested in mice for running endurance. In sedentary mice, 4 weeks of AICAR treatment alone induced metabolic genes and enhanced running endurance by 44% [10]. However, it was also shown in the same study that AMPK activation and exercise training synergistically increased oxidative myofibers and running endurance in adult mice.

My recommendation for this research is to publish with minor revisions.

  1. Mangano, D. T.; Miao, Y.; Tudor, I. C.; Dietzel, C.; Investigators of the Multicenter Study of Perioperative Ischemia Research, G.; Ischemia, R.; Education, F., Post-reperfusion myocardial infarction: long-term survival improvement using adenosine regulation with acadesine. J Am Coll Cardiol 2006, 48, (1), 206-14.
  2. Van Den Neste, E.; Cazin, B.; Janssens, A.; Gonzalez-Barca, E.; Terol, M. J.; Levy, V.; Perez de Oteyza, J.; Zachee, P.; Saunders, A.; de Frias, M.; Campas, C., Acadesine for patients with relapsed/refractory chronic lymphocytic leukemia (CLL): a multicenter phase I/II study. Cancer Chemother Pharmacol 2013, 71, (3), 581-91.
  3. Dixon, R.; Gourzis, J.; McDermott, D.; Fujitaki, J.; Dewland, P.; Gruber, H., AICA-riboside: safety, tolerance, and pharmacokinetics of a novel adenosine-regulating agent. J Clin Pharmacol 1991, 31, (4), 342-7.
  4. Nair, A. B.; Jacob, S., A simple practice guide for dose conversion between animals and human. J Basic Clin Pharm 2016, 7, (2), 27-31.
  5. Mangano, D. T., Effects of acadesine on myocardial infarction, stroke, and death following surgery. A meta-analysis of the 5 international randomized trials. The Multicenter Study of Perioperative Ischemia (McSPI) Research Group. JAMA 1997, 277, (4), 325-32.
  6. Drew, B. G.; Kingwell, B. A., Acadesine, an adenosine-regulating agent with the potential for widespread indications. Expert Opin Pharmacother 2008, 9, (12), 2137-44.
  7. Mahaffey, K. W.; Puma, J. A.; Barbagelata, N. A.; DiCarli, M. F.; Leesar, M. A.; Browne, K. F.; Eisenberg, P. R.; Bolli, R.; Casas, A. C.; Molina-Viamonte, V.; Orlandi, C.; Blevins, R.; Gibbons, R. J.; Califf, R. M.; Granger, C. B., Adenosine as an adjunct to thrombolytic therapy for acute myocardial infarction: results of a multicenter, randomized, placebo-controlled trial: the Acute Myocardial Infarction STudy of ADenosine (AMISTAD) trial. J Am Coll Cardiol 1999, 34, (6), 1711-20.
  8. Pokrywka, A.; Cholbinski, P.; Kaliszewski, P.; Kowalczyk, K.; Konczak, D.; Zembron-Lacny, A., Metabolic modulators of the exercise response: doping control analysis of an agonist of the peroxisome proliferator-activated receptor delta (GW501516) and 5-aminoimidazole-4-carboxamide ribonucleotide (AICAR). J Physiol Pharmacol 2014, 65, (4), 469-76.
  9. Visnjic, D.; Lalic, H.; Dembitz, V.; Tomic, B.; Smoljo, T., AICAr, a Widely Used AMPK Activator with Important AMPK-Independent Effects: A Systematic Review. Cells 2021, 10, (5).
  10. Narkar, V. A.; Downes, M.; Yu, R. T.; Embler, E.; Wang, Y. X.; Banayo, E.; Mihaylova, M. M.; Nelson, M. C.; Zou, Y.; Juguilon, H.; Kang, H.; Shaw, R. J.; Evans, R. M., AMPK and PPARdelta agonists are exercise mimetics. Cell 2008, 134, (3), 405-415.
  11. Xu, X.; Wang, W.; Wang, Z.; Lv, J.; Xu, X.; Xu, J.; Yang, J.; Zhu, X.; Lu, Y.; Duan, W.; Huang, X.; Wang, J.; Zhou, J.; Shen, X., DW14006 as a Direct AMPKalpha Activator Ameliorates Diabetic Peripheral Neuropathy in Mice. Diabetes 2020, 69, (9), 1974-1988.
  12. Chandrasekaran, K.; Salimian, M.; Konduru, S. R.; Choi, J.; Kumar, P.; Long, A.; Klimova, N.; Ho, C. Y.; Kristian, T.; Russell, J. W., Overexpression of Sirtuin 1 protein in neurons prevents and reverses experimental diabetic neuropathy. Brain 2019, 142, (12), 3737-3752.
  13. Chandrasekaran, K.; Najimi, N.; Sagi, A. R.; Yarlagadda, S.; Salimian, M.; Arvas, M. I.; Hedayat, A. F.; Kevas, Y.; Kadakia, A.; Russell, J. W., NAD(+) Precursors Repair Mitochondrial Function in Diabetes and Prevent Experimental Diabetic Neuropathy. Int J Mol Sci 2022, 23, (9).
  14. Chandrasekaran, K.; Najimi, N.; Sagi, A. R.; Yarlagadda, S.; Salimian, M.; Arvas, M. I.; Hedayat, A. F.; Kevas, Y.; Kadakia, A.; Kristian, T.; Russell, J. W., NAD(+) Precursors Reverse Experimental Diabetic Neuropathy in Mice. Int J Mol Sci 2024, 25, (2).
  15. Chaudhry, V.; Cornblath, D. R.; Mellits, E. D.; Avila, O.; Freimer, M. L.; Glass, J. D.; Reim, J.; Ronnett, G. V.; Quaskey, S. A.; Kuncl, R. W., Inter- and intra-examiner reliability of nerve conduction measurements in normal subjects. Ann. Neurol 1991, 30, (6), 841-843.
  16. Biessels, G. J.; Bril, V.; Calcutt, N. A.; Cameron, N. E.; Cotter, M. A.; Dobrowsky, R.; Feldman, E. L.; Fernyhough, P.; Jakobsen, J.; Malik, R. A.; Mizisin, A. P.; Oates, P. J.; Obrosova, I. G.; Pop-Busui, R.; Russell, J. W.; Sima, A. A.; Stevens, M. J.; Schmidt, R. E.; Tesfaye, S.; Veves, A.; Vinik, A. I.; Wright, D. E.; Yagihashi, S.; Yorek, M. A.; Ziegler, D.; Zochodne, D. W., Phenotyping animal models of diabetic neuropathy: a consensus statement of the diabetic neuropathy study group of the EASD (Neurodiab). J Peripher. Nerv. Syst 2014, 19, (2), 77-87.
  17. O'Brien, J.; Niehaus, P.; Chang, K.; Remark, J.; Barrett, J.; Dasgupta, A.; Adenegan, M.; Salimian, M.; Kevas, Y.; Chandrasekaran, K.; Kristian, T.; Chellappan, R.; Rubin, S.; Kiemen, A.; Lu, C. P.; Russell, J. W.; Ho, C. Y., Skin keratinocyte-derived SIRT1 and BDNF modulate mechanical allodynia in mouse models of diabetic neuropathy. bioRxiv 2024.
  18. O'Brien, P. D.; Guo, K.; Eid, S. A.; Rumora, A. E.; Hinder, L. M.; Hayes, J. M.; Mendelson, F. E.; Hur, J.; Feldman, E. L., Integrated lipidomic and transcriptomic analyses identify altered nerve triglycerides in mouse models of prediabetes and type 2 diabetes. Dis Model Mech 2020, 13, (2).

Round 2

Reviewer 1 Report

Comments and Suggestions for Authors

None

Author Response

This reviewer has no comments and suggestion, and we appreciate the input.

Reviewer 2 Report

Comments and Suggestions for Authors

The authors have made some additions in the paper, providing more information that was needed.

They reply correctly to the previous reviewer comments. However, they are not able to provide information on some important parameters, such as the amplitude of CAMPs and SNAPs, and the number of innervated Meissner's corpuscles in the pawpads. For the missing information on potential amplitudes they recall the EASD consensus in 2014, but they did not apply several other recommendations of that consensus, such as serial testing in all the studies, Meissner's corpuscles assessment, or nerve morphometry. They should indicate such limitations in the performed study. For exampla if the potential amplitudes are so variable in your laboratory, it is a methodological shortcoming that may be corrected by more strict technical training.

Author Response

Response to reviewer 2 is enclosed, appreciate the constructive suggestions.
